# Modular assembly of an artificially concise biocatalytic cascade for the manufacture of phenethylisoquinoline alkaloids

Yue Gao[1,3], Fei Li[1,3], Zhengshan Luo[1,3], Zhiwei Deng [1], Yan Zhang[2], Zhenbo Yuan [1], Changmei Liu[1] & Yijian Rao [1] ✉

Plant-derived alkaloids are an important class of pharmaceuticals. However, they still rely on phytoextraction to meet their diverse market demands. Since multistep biocatalytic cascades have begun to revolutionize the manufacture of natural or unnatural products, to address the synthetic challenges of alkaloids, herein we establish an artificially concise four-enzyme biocatalytic cascade with avoiding plant-derived P450 modification for synthesizing phenethylisoquinoline alkaloids (PEIAs) after enzyme discovery and enzyme engineering. Efficient biosynthesis of diverse natural and unnatural PEIAs is realized from readily available substrates. Most importantly, the scale-up preparation of the colchicine precursor (S)-autumnaline with a high titer is achieved after replacing the rate-limiting O-methylation by the plug-and-play strategy. This study not only streamlines future engineering endeavors for colchicine biosynthesis, but also provides a paradigm for constructing more artificial biocatalytic cascades for the manufacture of diverse alkaloids through synthetic biology.

Plant-derived tetrahydroisoquinoline (THIQ) alkaloids have widely been used to treat human diseases[1–5]. Based on their precursor scaffold structures[6], they can be divided into benzylisoquinoline, phenethylisoquinoline (Fig. 1a), ipecac, and amaryllidaceae alkaloid. Due to their structural complexity, they are still mainly obtained through phytoextraction since chemical synthesis is usually not commercially competitive[7–9]. However, their contents in medicinal plants are susceptible to weather, climate change, pest and locations, risking their market supply[1,10]. To address these issues, microorganisms have been engineered to achieve de novo biosynthesis of several well-known benzylisoquinoline alkaloids (BIAs)[11–13], such as opioids[1], guattegaumerine[10], and noscapine[2]. Nevertheless, the biosynthesis of other THIQ alkaloids in engineered microbial strains is still largely unknown, owing to the lack of elucidation of their biosynthetic pathways.

Recently, the biosynthetic pathway of colchicine in *Gloriosa superba* has been elucidated (Fig. 1b)[14,15]. This makes it possible to biosynthesize colchicine or other phenethylisoquinoline alkaloid (PEIA) drugs (Fig. 1a), such as bulbocodine, melanthiodine, kreysigine and harringtonine[16,17], in microorganisms (Fig. 1b). However, similar to BIAs, efficient synthesis of PEIAs in microorganisms is challenging because of their intricate biosynthetic pathway and the difficulty in expressing plant-derived P450 enzymes[1,13,18,19]. This leads to the low titer of the final products or intermediates[20–22]. To overcome these limitations, the multi-enzyme cascade reaction, which is an enzymatic procedure involving two or more steps for producing valuable chemical compounds from readily available (commercially available/easily accessible) precursor[23], maybe a powerful tool to sustainably synthesize valuable natural or unnatural products[24,25]. For instance, the HIV drug islatravir is efficiently

[1]Key Laboratory of Carbohydrate Chemistry and Biotechnology, Ministry of Education, School of Biotechnology, Jiangnan University, Wuxi 214122, PR China. [2]School of Life Sciences and Health Engineering, Jiangnan University, Wuxi 214122, PR China. [3]These authors contributed equally: Yue Gao, Fei Li, Zhengshan Luo. ✉e-mail: raoyijian@jiangnan.edu.cn

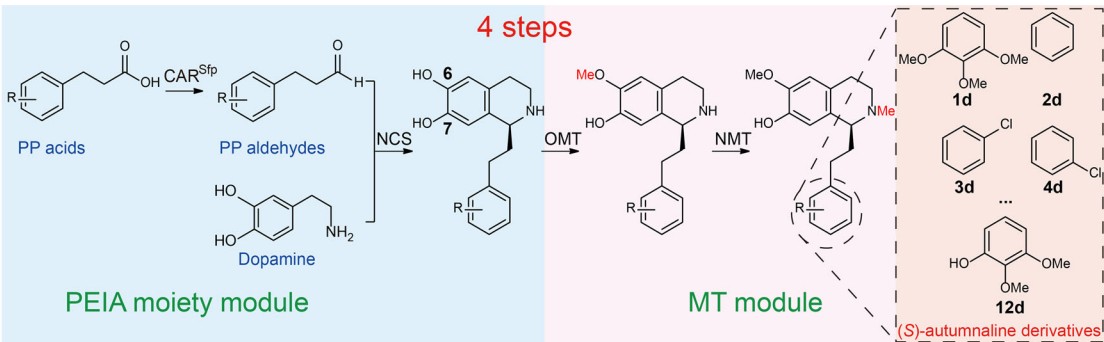

**Fig. 1 | Design of an artificially concise multi-enzyme cascade for the biosynthesis of PEIAs. a** Representative biologically active compounds containing the PEIA moiety. The PEIA moiety is highlighted in red. **b** Natural biosynthetic pathway of (S)-autumnaline in G. superba with more than 7 steps. **c** Construction of an artificial multi-enzyme cascade with the PEIA moiety module and the MT module

for the biosynthesis of (S)-autumnaline and its derivatives. The advantages of this cascade are indicated. CAR carboxylic acid reductase, Sfp phosphopantetheine transferase, NCS norcoclaurine synthase, OMT O-methyl transferase, NMT N-methyl transferase, PP acids phenylpropionic acids, PP aldehydes phenylpropionic aldehydes.

biosynthesized by a nine-enzyme biocatalytic cascade, which overcomes an unfavorable equilibrium, eliminates the accumulation of inhibitory and unstable intermediates, and avoids the purification of intermediates compared with chemical synthesis[25]. Furthermore, combined with the strategy of one-pot multi-step and metabolic engineering, which will circumvent the issues of the burden of protein expression, deficiency of cofactor and side reactions in a single cell[26], the titer of target products could be improved. The plug-and-play strategy could also be integrated to replace the rate-limiting enzymes involved in the synthesis of specific target products[27,28]. Most importantly, the stereo- and chemoselectivity of natural products (NPs) and drugs could also

be realized by enzymes. These appealing advantages of multistep biocatalytic cascades have revolutionized the manufacture of natural or unnatural products[25,29]. We therefore aimed to address the synthetic challenges of natural and unnatural PEIAs by rationally designing an artificially concise multi-enzyme cascade with easily available compounds as the substrates (Fig. 1c), which has never been reported before.

In this study, after enzyme discovery and enzyme engineering, we develop an artificial four-enzyme biocatalytic cascade to efficiently biosynthesize the colchicine precursor (S)-autumnaline and its various derivatives from readily available substrates in Escherichia coli (E. coli) (Fig. 1c). This method has a much shorter synthetic pathway and avoids

the modification by P450 CYP75A109 (Fig. 1b). After optimizing the process of four enzymes through metabolic engineering, various (*S*)-autumnaline derivatives are efficiently produced. Furthermore, the scale-up preparation of (*S*)-autumnaline with a high titer is realized when the rate-limiting *O*-methylation is replaced by the plug-and-play strategy (Fig. 1c). Therefore, this work provides a platform to streamline future engineering endeavors for the production of colchicine and other complex PEIAs.

## Results

### Design of an artificially concise cascade to biosynthesize (*S*)-autumnaline and its derivatives in vitro

To construct a platform for efficient biosynthesis of (*S*)-autumnaline and its unnatural derivatives with easily available compounds as the substrates (Fig. 1c), we analyzed the biosynthetic pathway of (*S*)-autumnaline based on that of colchicine (Fig. 1b). It suggests that they could be produced from the PEIA moiety derived from phenylpropionic aldehydes (PP aldehydes) and dopamine, followed by methylation (Figs. 1c, 2a, b). Thus, their biosynthetic pathway could be divided into two modules: the PEIA moiety module and the methyl transfer (MT) module (Fig. 1c). The PEIA backbone could be produced from PP aldehyde and dopamine by a norcoclaurine synthase (NCS)[30,31]. Considering that PP aldehydes are easily converted to alcohols in microorganisms[32], a carboxylic acid reductase (CAR) was introduced to convert phenylpropionic acids (PP acids) to the desired aldehydes in situ. Together with the following MT module using an *O*-methyltransferase and a *N*-methyltransferase, an artificially concise four-enzyme cascade (4 steps) was designed for the synthesis of (*S*)-autumnaline and its various derivatives (Fig. 1c), which is much shorter than the natural biosynthetic pathway (>7 steps), particularly avoiding the modification by P450 CYP75A109 (Fig. 1b).

Next, we used the compound 3-(3,4,5-trimethoxyphenyl) propanoic acid (**1**) as the model substrate and tested whether this artificial cascade could afford (*S*)-autumnaline derivative (*S*)−**1d**. At first, NCSs from *Thalictrum flavum* (*Tf*NCS)[30] and *Coptis japonica* (*Cj*NCS)[31] were selected to determine their feasibility to synthesize the PEIA moiety since the related enzyme responsible for Pictet-Spengler condensation reaction from *G. superba* is not reported[14]. As for CAR, *Tp*CAR from *Tsukamurella paurometabola* was chosen due to its broad substrate scope[33]. It clearly shows that (*S*)−**1b** was efficiently synthesized without the accumulation of intermediate **1a** (Fig. 2a) when **1** and dopamine were incubated with purified *Tf*NCS (or *Cj*NCS) and *Tp*CAR (Supplementary Fig. 1a). Considering a better expression (Supplementary Fig. 1b), *Tf*NCS was used for further investigation.

Then, we utilized (*S*)−**1b** as the substrate to screen the suitable OMT and NMT enzymes for the assembly of the MT module. *Rn*COMT from *Rattus norvegicus*[34] and *Gs*OMT1 from *G. superba*[10] were first studied to verify whether (*S*)−**1b** was able to be converted into (*S*)−**1c**. The results showed that both of them could realize the transformation (Supplementary Fig. 2a, b), but the enzymatic activity of *Rn*COMT towards (*S*)−**1b** was much better than that of *Gs*OMT1 (Supplementary Fig. 2c). Thus, *Rn*COMT was selected. Next, based on the biosynthetic pathway of colchicine[14], *Gs*NMT from *G. superba* was selected with truncation of the first 36 amino acids (*Gs*NMTt), which greatly enhanced the catalytic activity towards its native substrate (*S*)-**S1c** in *Nicotiana benthamiana*[14], to analyze its ability to transform (*S*)−**1c** into (*S*)−**1d**. However, (*S*)−**1d** could not be afforded when (*S*)−**1b** was incubated with purified *Rn*COMT and *Gs*NMTt (Fig. 2b), even though *Gs*NMTt could be well expressed (Supplementary Fig. 3a). Unexpectedly, its native substrate (*S*)-**S1c** also showed no transformation (Supplementary Fig. 3b, c). These data demonstrate that *Gs*NMTt has no enzymatic activity towards (*S*)−**1c** and (*S*)-**S1c**. The reason could be that the functional expression of plant-derived enzyme *Gs*NMTt may require specific post-translation modification or be organelle-specific[20]. Then, more NMTs from different sources including CNMT

from *C. japonica*[35], TNMT from *Glaucium flavum*[36] and PavNMT from *T. flavum*[37] were screened to realize the *N*-methylation of (*S*)−**1c**. To our delight, CNMT showed a very weak catalytic activity towards (*S*)−**1c** (Fig. 2b), suggesting that structure-guided engineering of CNMT is necessary to improve its catalytic activity.

### Structure-guided engineering of CNMT to improve its catalytic activity towards PEIA

To improve the catalytic activity of CNMT for the *N*-methylation of (*S*)−**1c** (Fig. 2b), structure-guided engineering of CNMT was performed. The substrate (*S*)−**1c** was docked into crystal structure of CNMT (PDB: 6GKY). Then, amino acids within 4.0 Å of CNMT surrounding the trimethoxyphenyl group of (*S*)−**1c**, including L88, M91, N92, W297, W329, F332, and C333 (Supplementary Fig. 4), were selected to conduct alanine scanning mutagenesis. All variants, except CNMT-W297A (Supplementary Fig. 5), could be purified. It shows that variants CNMT-L88A, CNMT-N92A, and CNMT-F332A exhibited increased catalytic activity of CNMT by 1.43, 5.20 and 7.06 times, respectively (Fig. 2c). Here, the catalytic activity of CNMT towards (*S*)−**1c** was reflected by detecting the titer of (*S*)−**1d**. Based on the results of alanine scanning mutagenesis, semi-rational design was conducted to further enhance the catalytic activity of CNMT. To establish potential hydrogen bonds between CNMT and the substrate (*S*)−**1c**, both L88 and F332 were mutated to serine. In addition, to obtain a better substrate binding conformation for substrate (*S*)−**1c**, N92 was mutated to valine. Finally, to investigate the steric effects of CNMT on the substrate (*S*)−**1c**, L88 was mutated to glycine and valine, and F332 was mutated to valine. The results showed that only CNMT-L88V further increased the catalytic activity of CNMT by 2.25 times (Fig. 2c). Thus, combinatorial mutagenesis was adopted to construct different variants, showing that CNMT[N92A/F332A] (named as CNMT\*) greatly improved the catalytic activity, up to 17.26 times (Fig. 2c). According to the results of molecular docking, N92A/F332A could cause a downward shift of (*S*)−**1c** compared with wild-type CNMT (CNMT[WT]) (Supplementary Fig. 4a), resulting in the formation of a new hydrogen bond interaction between (*S*)−**1c** and N294 at a distance of 3.0 Å, which is 4.7 Å in CNMT[WT] (Supplementary Fig. 4a). Moreover, a larger substrate pocket is observed (Supplementary Fig. 6), allowing a better substrate binding. It suggests that this new conformation improves the catalytic activity of CNMT\* towards (*S*)−**1c**.

With the availability of CNMT\*, we next evaluated the feasibility of the whole cascade reaction via a one-pot two-step process in vitro, in which the steps catalyzed by the PEIA moiety module and by the OMT module are performed in sequential mode, obviating the need for purification of the intermediates and avoiding the side reaction caused by *Rn*COMT towards dopamine[34]. It showed that (*S*)−**1d** was efficiently prepared with >98% *ee* (Fig. 2d, Supplementary Fig. 7), when **1** and dopamine were incubated with purified *Tp*CAR, *Tf*NCS, *Rn*COMT, and CNMT\* in vitro (Supplementary Fig. 1). Moreover, it is also found that (*S*)−**1b** should be first *O*-methylated by *Rn*COMT and then undergoes *N*-methylation by CNMT\* to synthesize (*S*)−**1d** (Supplementary Fig. 8). Based on the above results, we conclude that an artificially concise four-enzyme cascade reaction is successfully constructed to synthesize (*S*)−**1d**.

### Construction of the PEIA moiety module in engineered strain IAA

To enable the large-scale production of PEIAs and circumvent the issues including elaborate protein purification and expensive cofactors that are caused by in vitro reconstitution, the whole-cell biocatalysis (in vivo) was applied. Firstly, we separately introduced the PEIA moiety module and the MT module into *E. coli* BL21 (*DE3*) to produce PEIAs in a one-pot two-step process, where the strains containing the PEIA moiety module and the MT module are added consecutively (Fig. 2a, b). Meanwhile, to impair the rapid conversion of the intermediate aldehyde **1a** into the corresponding alcohol in *E. coli*, seven

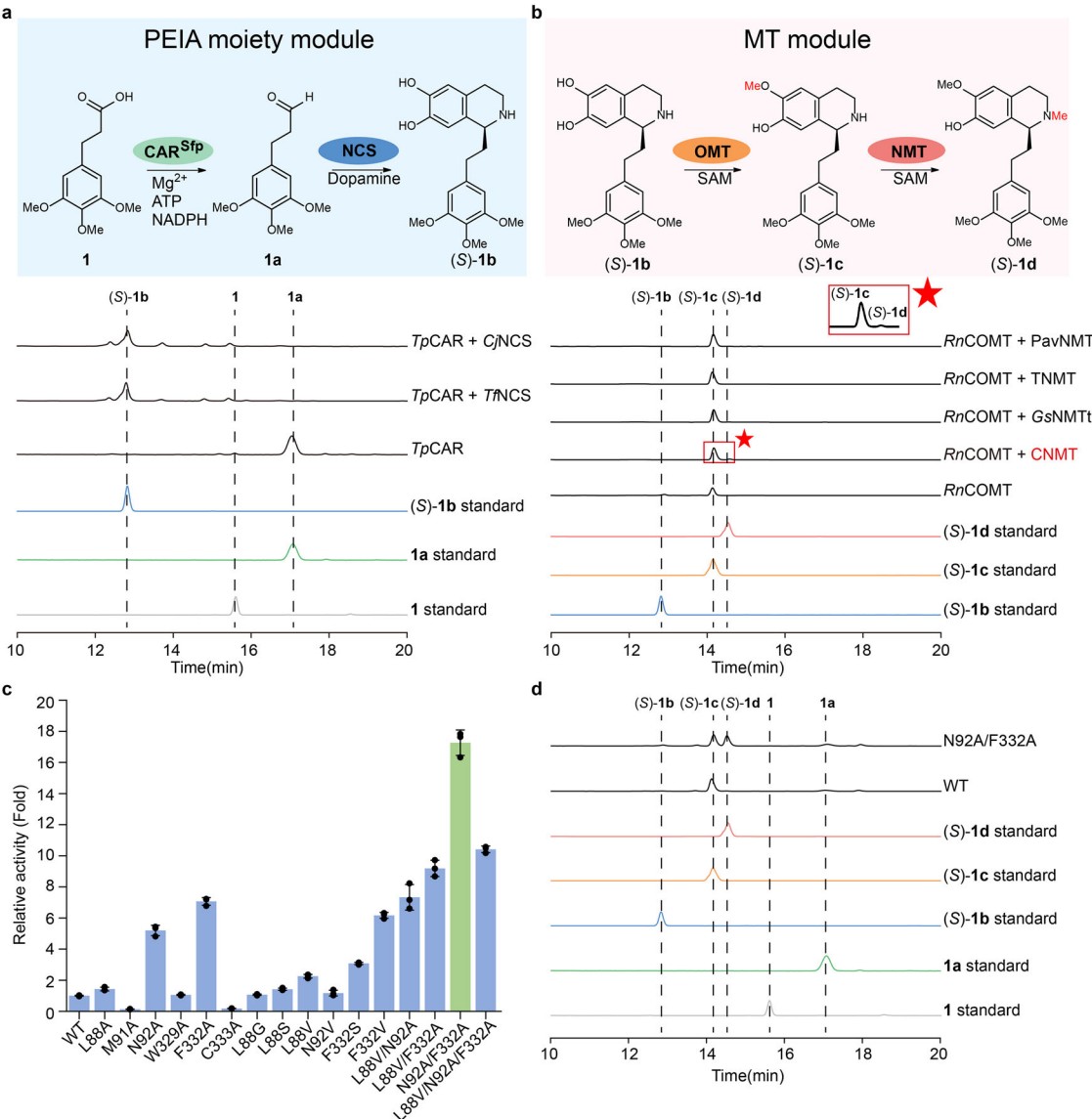

**Fig. 2 | Establishment of a multi-enzyme cascade reaction to efficiently synthesize (S)-autumnaline derivatives in vitro. a** The PEIA moiety module with CAR and NCS. CAR requires $Mg^{2+}$, ATP, and NADPH for its activity. CAR activated by Sfp is named as $CAR^{Sfp}$ (*holo*-CAR). The catalytic activity of *Tp*CAR, *Cj*NCS and *Tf*NCS was analyzed by high-performance liquid chromatography (HPLC). The intermediate **1a** and (S)-**1b** in the PEIA moiety module were detected. **b** The MT module with OMT and NMT, which require SAM as the methyl donor. The catalytic activity of *Rn*COMT and NMT from various sources was determined by HPLC analysis. The intermediate (S)-**1c** and product (S)-**1d** in the MT module were detected. The red star represents a partial enlargement view of trace (S)-**1d**. **c** Relative activity of alanine-scanning mutants and combinational mutants of selected residues of CNMT towards (S)-**1c**. The catalytic activity of CNMT was assessed by detecting the titer of (S)-**1d** and the relative activity was determined by the ratio of the (S)-**1d** titer of CNMT mutants to that of CNMT$^{WT}$. The mutant with the highest catalytic activity is highlighted by green column. **d** HPLC analysis of the biocatalytic cascade reactions containing the PEIA moiety module and the MT module in a one-pot two-step process in vitro to produce (S)-**1d** from **1** and dopamine in vitro. In the PEIA moiety module, substrate **1**, dopamine, enzymes, and cofactors were added to the reaction vessel and reacted for 4 h. The reaction solution was boiled and centrifuged to collect the supernatant. In the MT module, enzymes and co-factors were added to the resulting supernatant and incubated for 4 h. All data is presented as mean value of three independent experiments and the error bars indicate ± sd. Source data are provided as a Source Data file. ATP adenosine triphosphate, SAM *S*-adenosylmethionine.

genes encoding aldo-keto reductases (AKR) (*dkgB*, *yeaE* and *dkgA*), alcohol dehydrogenases (ADH) (*yqhD*, *yahK* and *yjgB*) and a transcription factor *yqhC* were knocked out (Fig. 3a)[32,38], giving the engineered improvement of aldehyde accumulation (IAA) strain. Then, nine strains containing different combinations of plasmids were constructed to optimize the PEIA moiety module for the preparation of (S)−**1b** (Fig. 3b). It showed that, although all combinations produced (S)−**1b**, the strain M1G exhibited the highest yield, surpassing that of WT strain transformed with the same plasmid as M1G (Fig. 3c). This finding well supports the notion that knock out of the above seven genes indeed inhibits the production of by-product 3-(3,4,5-

trimethoxyphenyl) propan−1-ol (**1a'**), thereby increasing the titer of (S)−**1b**. To minimize side reactions caused by residual dopamine catalyzed by the following MT module[34], excessive substrate **1** was used. Within 8 h, 7.5 mM **1** and 5 mM dopamine were efficiently converted to 4.48 mM (S)−**1b** by M1G with a yield of 89.60%. Although the titer of **1a'** was greatly decreased by knocking out the above seven genes, a small amount of **1a'** was still produced from **1a** catalyzed by other endogenous enzymes, which may be inevitable[32]. Furthermore, it shows that most of (S)−**1b** were mainly found in the supernatant (Fig. 3d, Supplementary Fig. 9), which will be a great benefit to synthesize downstream products by the one-pot multi-step strategy.

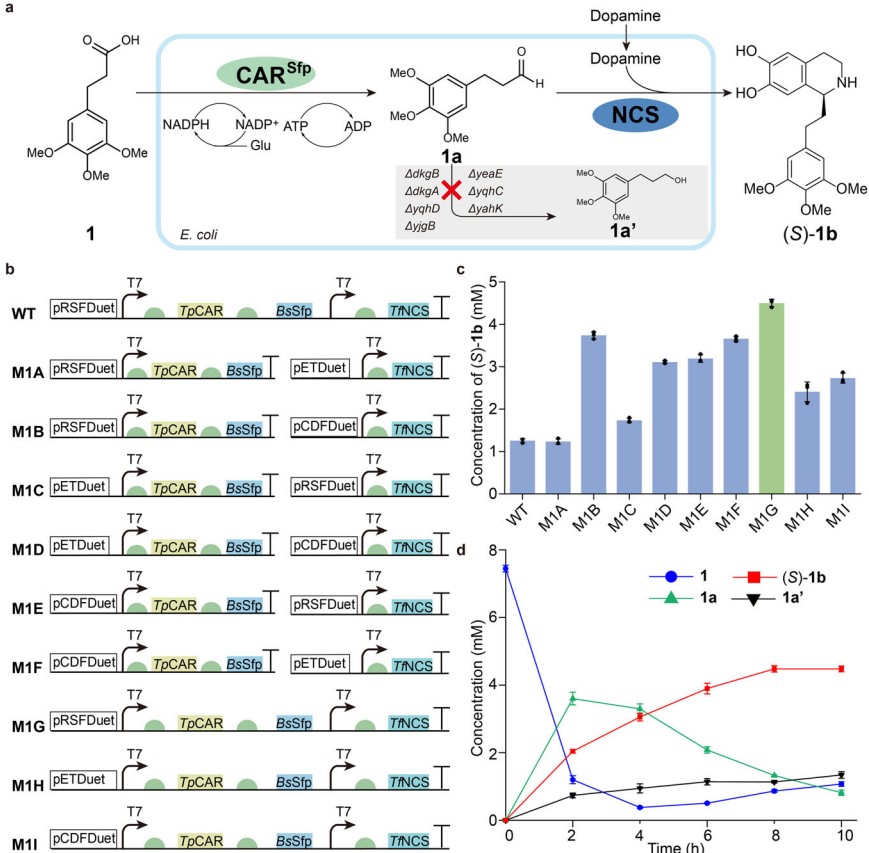

**Fig. 3 | Biosynthesis of (S)-1b in engineered strain IAA. a** Illustration of the designed biosynthetic pathway of the PEIA moiety with blocking the production of by-product **1a'** in *E. coli*. The red cross indicates that several genes encoding ADH (*yqhD, yahK, yjgB*), AKR (*dkgB, yeaE, dkgA*), and a transcription factor (*yqhC*) were knocked out. **b** Construction of nine engineered strains expressing *Tp*CAR, *Bs*Sfp and *Tf*NCS. The green semicircle indicates the ribosome binding site. **c** The titer of (S)-**1b** synthesized by engineered strains. 5 mM substrate **1** and 5 mM dopamine were used to produce (S)-**1b** in HEPES buffer (pH 7.5) by engineered strains containing the PEIA moiety module (*Tp*CAR, *Bs*Sfp, *Tf*NCS). The green column represents the strain M1G with the highest titer of (S)-**1b**. **d** Time course of converting 7.5 mM **1** and 5 mM dopamine to (S)-**1b** using whole-cell catalyst M1G. Blue line: **1**; green line: **1a**; red line: (S)-**1b**; black line: **1a'**. All data are presented as mean values of three independent experiments and the error bars indicate ±sd. Source data are provided as a Source Data file.

## Construction of the MT module in engineered strain IAA

Next, the MT module was introduced into another engineered strain IAA. To avoid the side effects on dopamine caused by the promiscuity of *Rn*COMT and the purification of the intermediate of (S)−**1b**, the one-pot two-step strategy (using two strains: the strain containing the PEIA module and the strain containing the MT module) was firstly applied to biosynthesize (S)−**1d** (Fig. 4a). It means that the supernatant of the PEIA moiety module reaction is directly added into the whole cell expressed *Rn*COMT and CNMT*. Meanwhile, to solve the issue of low content of SAM in *E. coli*[39], a SAM supply system was integrated into the MT module by coexpressing methionine adenosyltransferase (*Ec*MAT) from *E. coli* and *S*-adenosylhomocysteine hydrolase (*Mm*SAHH) from *Mus musculus*[40]. *Ec*MAT is responsible for synthesizing SAM from ATP and L-methionine while *Mm*SAHH can hydrolyze *S*-adenosylhomocysteine (SAH), an inhibitor of OMT[41]. Accordingly, it shows that (S)−**1d** could be afforded, but only with a total yield of 28.40%, even after optimizing the copy number of the plasmids (Fig. 4b). The reasons could be as follows: (i) excessive expression of exogenous proteins in a single cell might increase the metabolic burden of the strain (Supplementary Fig. 10)[26]; (ii) the ratio of *Rn*COMT and CNMT* likely affects the production of (S)−**1d**. Thus, it suggests that further division of the MT module into the OMT module and the NMT module could potentially enhance the production of (S)−**1d** (Fig. 4c), that is, one-pot three step manner.

Then, the OMT module containing *Rn*COMT, *Ec*MAT and *Mm*SAHH was constructed for the synthesis of (S)−**1c** (Fig. 4d), and three strains with different copies of plasmids were investigated for comparison (Supplementary Fig. 11a). All of them could afford (S)−**1c**, and the strain M2B (*Rn*COMT) exhibited the best performance (Supplementary Fig. 11b). Based on the time course (Fig. 4e), M2B (*Rn*COMT) could convert 92.86% of (S)−**1b** within 5 h after optimizing the concentration of M2B (*Rn*COMT) strain (Supplementary Fig. 11c, d). As for the NMT module, the same optimization method was employed (Fig. 4d). It is noteworthy that the pH of the reaction system after the step of the OMT module decreased to 6.5 due to the consumption of ATP, which seriously affected the catalytic efficiency of the NMT module. Thus, the pH of the supernatant was adjusted to 7.5 for the subsequent reaction. The results demonstrate that the strain M3B could produce 3.09 mM (S)−**1d** within 5 h, with a final total yield of 61.80% from **1** and dopamine (Fig. 4e, Supplementary Fig. 12). This is 2.18-fold higher than that of one-pot two-step method. Although this yield is still lower than the theoretical one, which may mainly come from the poor regioselectivity of *Rn*COMT with the formation of 7-methoxyl positional isomer of (S)−**1c** as the by-product[42], CNMT* only catalyzes the substrate with 6-OMe and 7-OH, thereby affording pure (S)−**1d** as determined by nuclear magnetic resonance (NMR) spectroscopy. In all, an artificially concise biocatalytic cascade is successfully developed for efficient biosynthesis of (S)-autumnaline

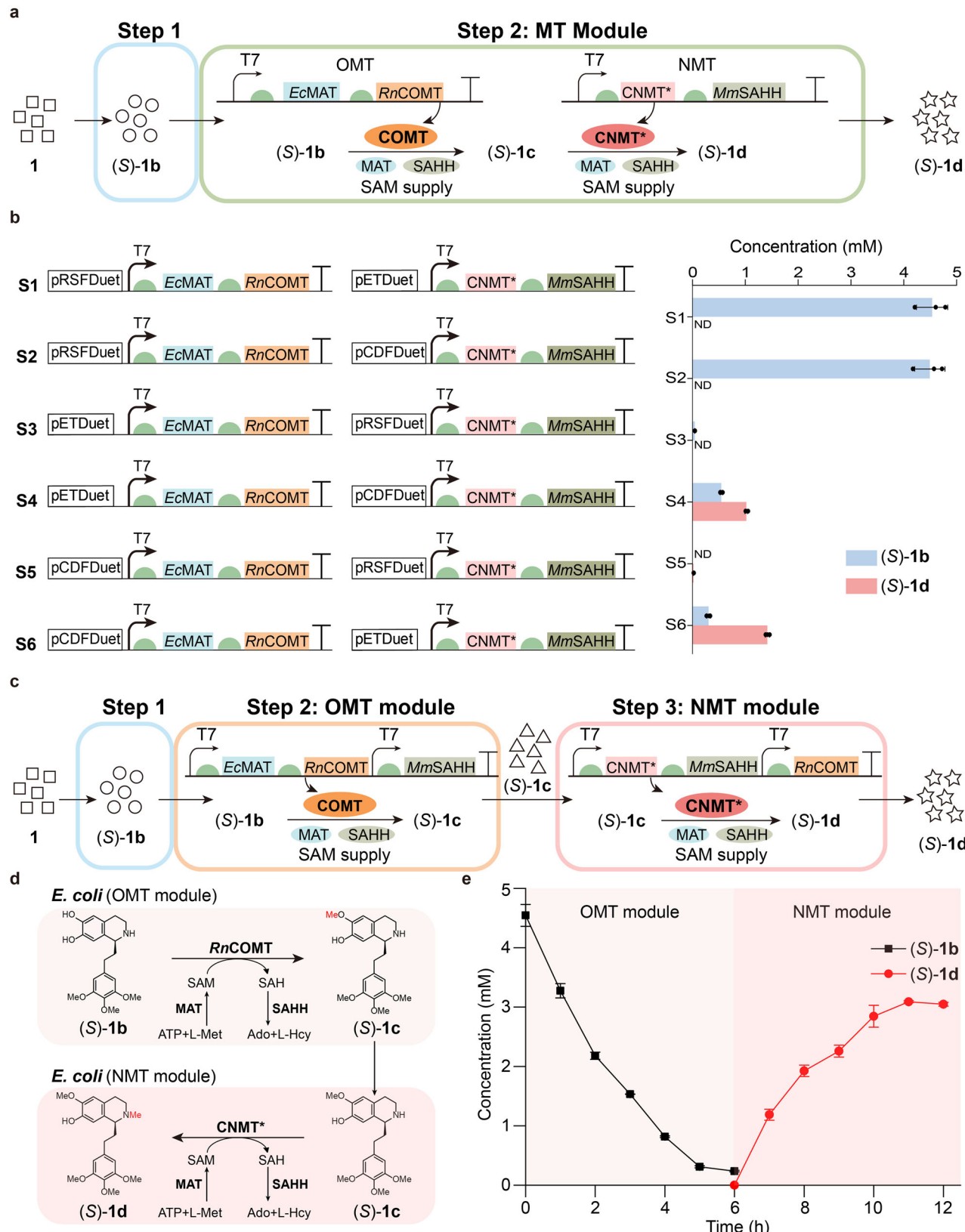

derivative (S)−**1d** from the easily available substrate by a multi-enzyme cascade.

## Substrate scope of the developed biocatalytic cascade

Next, we evaluated the ability of the above platform to synthesize (S)-autumnaline and other unnatural derivatives. At first, PP acids with

different types of single substituent group (**2**–**10**) on the different positions of benzene ring were studied (Fig. 5a). All of them could be well converted to the corresponding products ((S)-**2d** to (S)−**10d**) with yields ranging from 70.95% to 77.95% (Fig. 5b), which were better than that of (S)−**1d**. Then, the substrate with di-substituent groups (**11**) was tested, and the desired product (S)−**11d** could also be afforded in a

**Fig. 4 | Biosynthesis of (S)-1d in engineered strain IAA. a** Scheme of the designed biosynthetic pathway for the preparation of (S)-1d from substrate **1** and dopamine through the one-pot two-step strategy. Substrate **1** and dopamine were added to the suspension of the strain M1G and reacted for 8 h. M1G supernatant, obtained by centrifugation, was used to resuspend strains S1-S6 and reacted for 6 h. The square represents substrate **1**. The circle represents (S)-1b from the PEIA moiety module. The pentagram represents the final product (S)-1d. The SAM supplying system is constituted by MAT and SAHH. **b** The titer of (S)-1b, and (S)-1d synthesized by engineered strains (S1-S6) in a one-pot two-step process. ND: not detected. The copy number of pRSFDuet, pETDuet and pCDFDuet are >100, ~40 and 20-40, respectively. **c** Scheme of the designed biosynthetic pathway for the preparation of (S)-1d from substrate **1** and dopamine through the one-pot three-step strategy.

Substrate **1** and dopamine were added to the suspension of the strain M1G and reacted for 8 h. M1G supernatant, obtained by centrifugation, was used to resuspend strains M2A-M2C and reacted for 6 h. The resulting supernatant, adjusted to pH 7.5, was used to resuspend strains M3A-M3C and reacted for 6 h. The triangle represents (S)−1c from the OMT module. **d** Constructing the biosynthetic pathway of (S)−1d from (S)−1b through the OMT module and the NMT module. **e** Time course of the biotransformation of (S)−1b to (S)−1d using whole-cell catalyst M2B (RnCOMT) and M3B. The reaction process is shown in **c** and **d**. Black line: (S)−1b; red line: (S)−1d. All data is presented as mean value of three independent experiments and the error bars indicate ± sd. Source data are provided as a Source Data file. Ado adenosine, L-Hcy L-homocysteine.

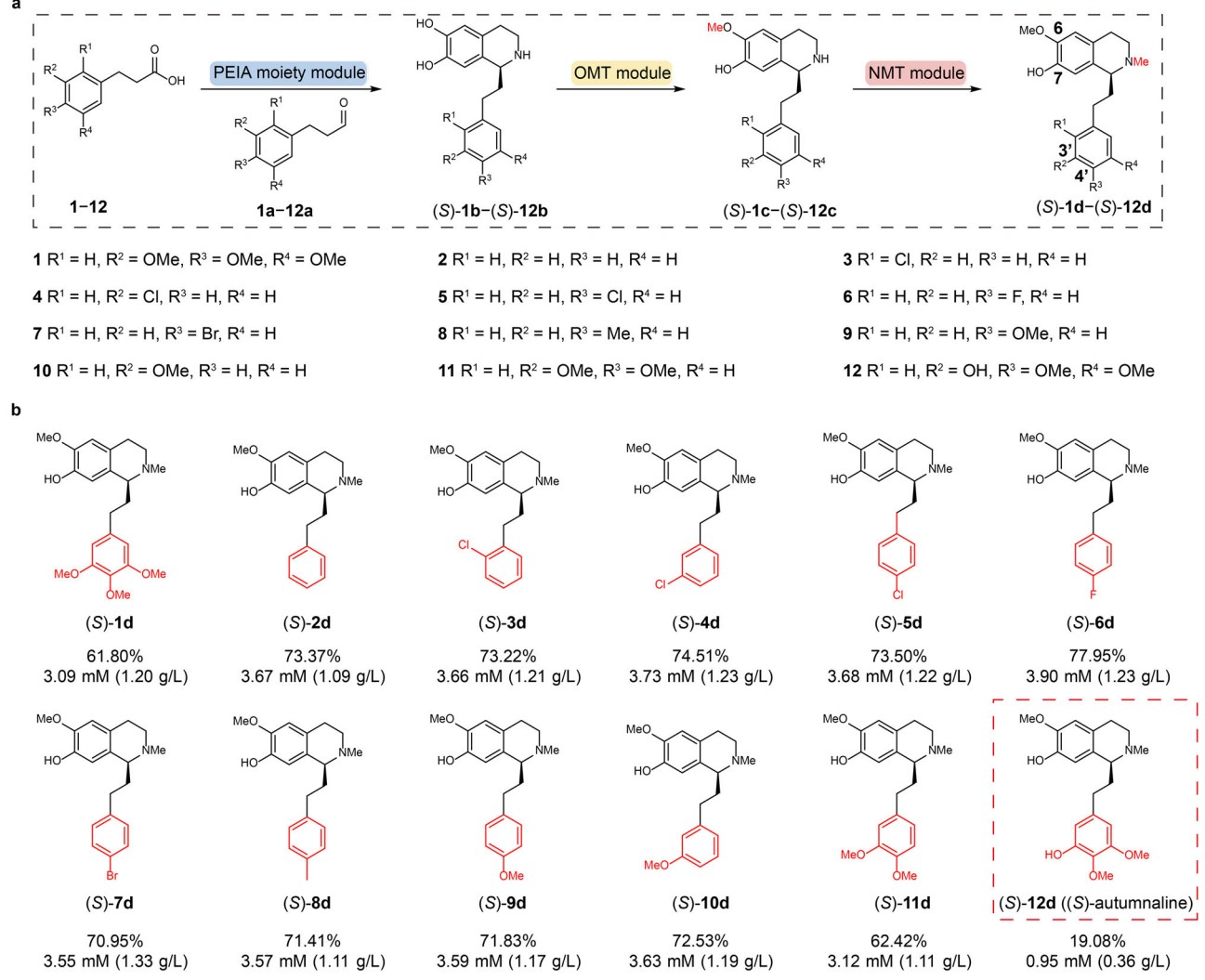

**Fig. 5 | Biosynthesis of (S)-autumnaline and its derivatives through the artificial multi-enzyme cascade reaction in a one-pot three-step process. a** Biosynthesis of (S)−1d−(S)−12d from substrate **1−12** and dopamine by the artificial multi-enzyme cascade reaction containing the PEIA moiety module, the OMT module and the NMT module in a one-pot three-step process. The procedure for synthesizing (S)-2d−(S)−12d is the same as the one for (S)−1d−(S)−12d. **b** The titer and yield of (S)−1d−(S)−12d synthesized by the artificial multi-enzyme cascade reaction are shown. All data are presented as mean value of three independent experiments. Source data are provided as a Source Data file.

similar yield of (S)−1d (62.42%) (Fig. 5b). Lastly, the substrate **12** used for the synthesis of the colchicine precursor (S)-autumnaline was investigated. Unfortunately, only 0.95 mM (S)−12d was produced with a yield of 19.08% (Fig. 5b). Through the analysis of HPLC results, it showed that most of (S)−12b could not be efficiently converted to (S)−12c by the OMT module (Supplementary Fig. 13), suggesting that RnCOMT is not an appropriate enzyme for (S)−12b. In general, the above result well demonstrates that the designed artificial biocatalytic cascade is a versatile and effective platform for preparing PEIAs from a broad range of PP acids.

### Improving the catalytic activity of GsOMT1 for scale-up preparation of (S)-autumnaline

Last, we aimed to overcome the low titer of (S)-autumnaline through the plug-and-play strategy. Then, original OMT from G. superba (GsOMT1) was selected[14]. It shows that its catalytic activity towards (S)

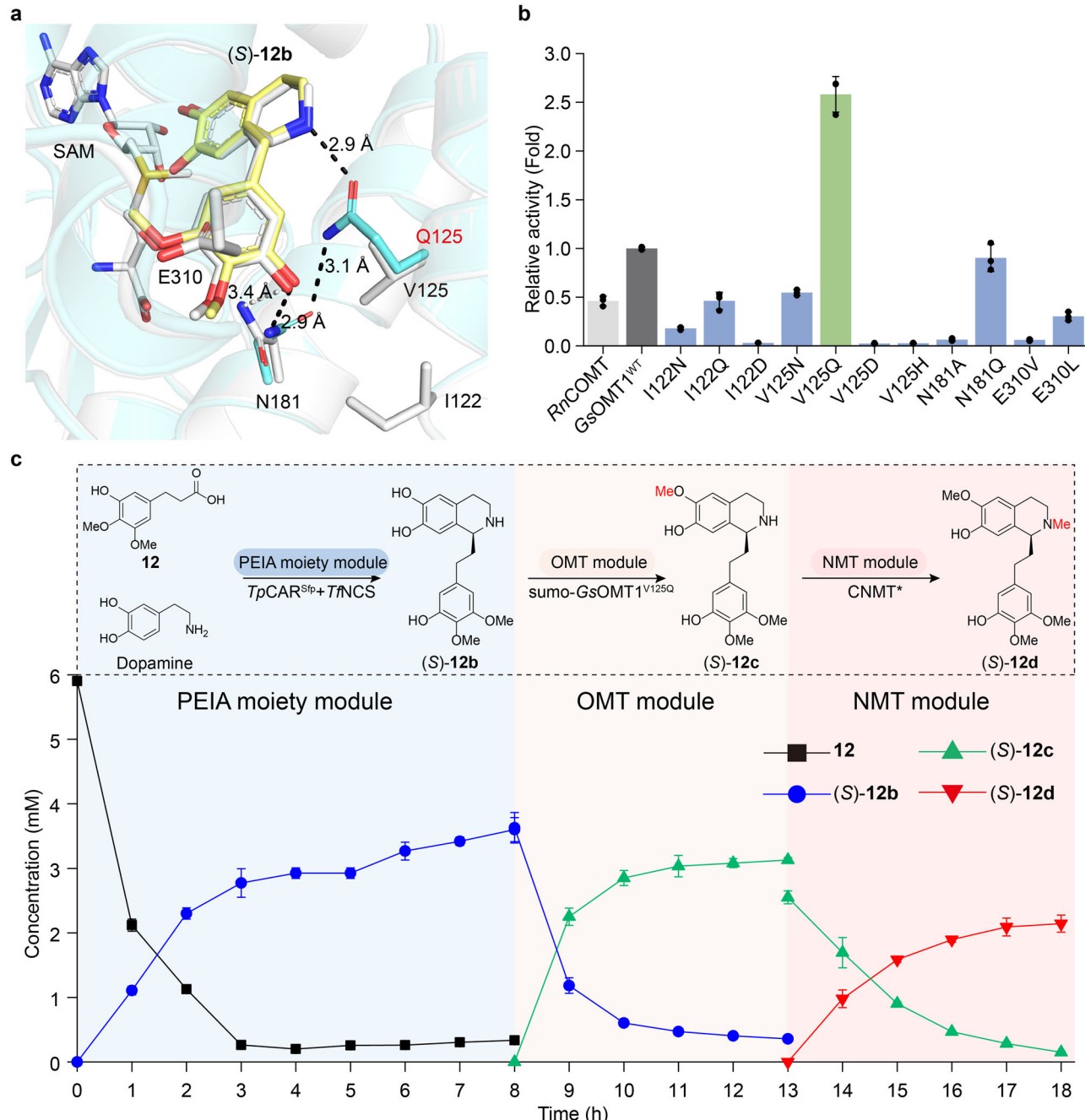

**Fig. 6 | Rational design of *Gs*OMT1 for efficient biosynthesis of (*S*)-autumnaline.**
**a** Comparison of the (*S*)−**12b** binding site in *Gs*OMT1^[WT] and *Gs*OMT1^[V125Q]. *Gs*OMT1^[WT] is displayed in white while *Gs*OMT1^[V125Q] is shown in cyan. (*S*)−**12b** in *Gs*OMT1^[V125Q] is colored by yellow. The selected residues in *Gs*OMT1^[WT] are indicated by white sticks. The mutated residues in *Gs*OMT1^[V125Q] are shown in cyan sticks. Oxygen, nitrogen and sulfur atom are shown in red, blue and yellow, respectively. The dash line indicates a hydrogen bond. **b** Relative catalytic activity of *Rn*COMT, *Gs*OMT1^[WT] and its mutants towards (*S*)−**12b**. The relative activity was determined by the ratio of the (*S*)−**12c** titer of *Gs*OMT1 mutants to that of *Gs*OMT1^[WT]. The mutant with the highest catalytic activity is highlighted by green column. **c** Time course of the

biotransformation of 6 mM **12** and 4 mM dopamine to (*S*)−**12d**. Substrate **12** and dopamine were added to the suspension of the strain M1G and reacted for 8 h. M1G supernatant, obtained by centrifugation, was used to resuspend the strain M2B (sumo-*Gs*OMT1^[V125Q]) and reacted for 6 h. The resulting supernatant, adjusted to pH 7.5, was used to resuspend the strain M3B and reacted for 6 h. Black line: **12**; blue line: (*S*)−**12b**; green line: (*S*)−**12c**; red line: (*S*)−**12d**. Due to high concentration of strain M2B (sumo-*Gs*OMT1^[V125Q]), a small amount of (*S*)−**12c** remains in the pellet after the OMT module. All data is presented as mean value of three independent experiments and the error bars indicate ± sd. Source data are provided as a Source Data file.

−**12b** is better than that of *Rn*COMT (Supplementary Fig. 14a). However, the titer of (*S*)−**12c** was still too low, with only 0.59 mM (*S*)−**12c** being produced from 1.60 mM (*S*)−**12b** (Supplementary Fig. 14b). Thus, rational protein engineering was performed to enhance its catalytic activity. Its structure was generated by AlphaFold2[43] and (*S*)−**12b** was docked into the resulting model (Fig. 6a, Supplementary Fig. 15). After

rational analysis, residues I122, V125, N181, and E310 surrounding 3′-hydroxy-4′,5′-diomethoxyphenethyl group of (*S*)−**12b** were selected for engineering. In order to enhance the binding affinity between the substrate (*S*)−**12b** and *Gs*OMT1, a potential hydrogen bonding network was constructed by mutating I122 and V125 to polar or charged amino acids. Meanwhile, to strengthen the hydrophobic interaction between

the substrate (S)−**12b** and *Gs*OMT1, E310 was mutated to nonpolar amino acids. Furthermore, to validate the hydrogen bonding interaction between the substrate (S)−**12b** and N181, N181 was mutated to alanine and glutamine. Accordingly, 11 mutants were constructed (Fig. 6b), and only the mutant V125Q could greatly increase the catalytic activity of *Gs*OMT1 (Fig. 6b, Supplementary Fig. 14b). This is well explained by the docking results of *Gs*OMT1$^{V125Q}$, demonstrating that V125Q induces the rotation of the side chain of N181 and leads to the formation of a new hydrogen bonding network among (S)−**12b**, N181 and Q125 (Fig. 6a). This is further supported by the enzyme kinetic parameters of *Gs*OMT1$^{WT}$ and *Gs*OMT1$^{V125Q}$ towards (S)−**12b** (Supplementary Fig. 16, Supplementary Table 1). *Gs*OMT1$^{V125Q}$ not only improves the binding affinity of (S)−**12b**, but also increases the conversion rate towards (S)−**12b**, leading to an increased $k_{cat}/K_m$ (Supplementary Table 1).

Next, the plug-and-play strategy was employed by replacing *Rn*COMT with *Gs*OMT1$^{V125Q}$ in the OMT module to construct a new strain M2B (*Gs*OMT1$^{V125Q}$). However, we found that the expression level of *Gs*OMT1$^{V125Q}$ in *E. coli* was low (Supplementary Fig. 17). To address this issue, a soluble sumo tag was introduced, resulting in a significant improvement in its expression (Supplementary Fig. 17). Then, the strain M2B (sumo-*Gs*OMT1$^{V125Q}$) was prepared to produce (S)−**12c**. After optimizing the concentration of M2B (sumo-*Gs*OMT1$^{V125Q}$) (Supplementary Fig. 18), 2.14 mM (S)-autumnaline was produced from 6 mM **12** and 4 mM dopamine with a moderate yield of 53.50% (Fig. 6c). Compared with the cascade reaction catalyzed by strains M1G, M2B (*Rn*COMT) and M3B, the yield of (S)-autumnaline was increased by 2.25 times. However, its yield is still low. The reason might involve the following several aspects: the oxidation of the dopamine by air or its consumption by the microorganisms, a part of (S)−**12c** remaining within the cell pellet due to the high density of M2B (sumo-*Gs*OMT1$^{V125Q}$) and the insufficient catalytic activity of *Gs*OMT1$^{V125Q}$ and CNMT* towards (S)−**12b** and (S)−**12c** (Fig. 6c), respectively. Therefore, in order to increase the titer of (S)-autumnaline, further protein engineering of *Gs*OMT1$^{V125Q}$ and CNMT* may be required.

Then, the scalability of the above three modules (M1G, M2B (sumo-*Gs*OMT1$^{V125Q}$) and M3B) was investigated. To our delight, 1.90 mM (709.06 mg L$^{-1}$) (S)-autumnaline could be obtained with a yield of 47.50% in a 300 mL reaction system, giving a similar yield to the small reaction system (5 mL). This result strongly indicates that the developed biocatalytic cascade has great potential for scale-up preparation of (S)-autumnaline, which will be a substantial benefit for the biosynthesis of drug colchicine.

## Discussion

In this work, combined with enzyme discovery, protein engineering and the one-pot multi-step strategy, an artificially concise multi-enzyme cascade is successfully established for efficient biosynthesis of diverse PEIAs from readily available substrates in *E. coli*. This circumvents the main challenges associated with heterologous reconstruction of intricate biosynthetic pathways starting from the long glycolytic pathway and the obstacles involved in expressing plant-derived P450 enzymes. More importantly, the colchicine precursor (S)-autumnaline is also efficiently produced by adopting the plug-and-play strategy to replace the rate-limiting *O*-methylation. This advancement enables the scale-up preparation of (S)-autumnaline in a good final titer.

Based on the results of this study and previous work[44,45], NCSs were observed to strictly recognize the amine moiety of the substrate and only accept the 3-hydroxyphenyl-2-ethylamine derivatives. However, NCSs could recognize a variety of aldehydes, which is supported by the results of substrate scope investigation. Although NCSs have great promiscuity for various aldehydes, the yields of final products from different aldehydes are affected by the promiscuity of the OMT module (Fig. 5b). Single substituent groups (**2**–**10**) on the different positions of benzene ring gave good yields of (S)-**2d**−(S)-**10d**, and the yields of di-,

tri-substituted products decreased slightly. However, as for the specific substrate **12**, this reaction system did not deliver a satisfactory result. This might be attributed to the similarities between the 6-OMe and 7-OH of (S)−**12c** and the 3′-OH and 4′-OMe of (S)−**12b** (Fig. 5a), which would occupy the binding pocket of *Rn*COMT to limit the production of (S)−**12c**. As for this kind of substrates, the discovery and engineering of new OMTs will be necessary for achieving high production of the desired PEIAs. To further increase the yields of final products, the strategies of enzyme engineering and metabolic engineering could be combined to optimize multi-enzyme cascade, and then effectively address the shortcomings of multi-stage biotransformation, such as several centrifugation and resuspension steps and ATP regeneration.

In this study, the established concise artificial pathway bypasses the reaction catalyzed by P450 CYP75A109, and shortens the biosynthetic steps for the biosynthesis of the colchicine precursor (S)-autumnaline and its derivatives, which could benefit the total biosynthesis of colchicine and its derivatives, or other PEIAs. However, as for the construction of specific skeletons, for instance, the core structure of colchicine, the following modification by P450 or other enzymes may be still required. In all, this work not only develops a platform for further biosynthesis of colchicine and other natural or unnatural PEIAs, but also paves the way to construct other artificial biocatalytic cascades for the manufacture of different alkaloids.

## Methods
### Chemicals and materials
The genes of *Tp*CAR (accession number: WP_013126039.1) from *T. paurometabola*, *Tf*NCS (accession number: ACO90248.1) from *T. flavum*, *Rn*COMT (accession number: NP_036663.1) from *R. norvegicus*, *Gs*OMT1 (accession number: QLI49050.1) from *G. superba*, CNMT (accession number: BAB71802.1) from *C. japonica*, *Gs*NMT (accession number: QLI49051.1) from *G. superba*, PavNMT (PDB: 5KN4) from *T. flavum*, TNMT (PDB: 6P3M) from *G. flavum* and *Mm*SAHH (accession number: NP_001291457.1) from *M. musculus* were synthesized and codon-optimized for *E. coli* BL21 (*DE3*) by Exsyn-Bio (Wuxi, China). The genes of *Bs*Sfp, GDH and *Ec*MAT were amplified from the genome of *Bacillus subtilis* and *E. coli*, respectively. The enzymes used for DNA ligation and Polymerase Chain Reaction (PCR) were purchased from Exsyn-bio (Wuxi, China) and Takara Ltd (Shanghai, China), respectively. Primers used for gene amplification were synthesized in Exsyn-Bio (Wuxi, China) and all constructed plasmids were sequenced by Genewiz (Suzhou, China). Supplementary Table 2 provides the information of the reagents and solvents used in this study.

### Plasmids and strains construction
Plasmids, strains, and primers used in this study are listed in Supplementary Tables 3, 4, and 5, respectively. Plasmids used in this study were constructed in the following protocol. The fragments for target gene and vectors (pET-21b (+), pRSFDuet−1, pETDuet−1 and pCDFDuet−1(Novagen)) were amplified by PCR using primers. The gene fragments, the vector fragments, ddH$_2$O and 2 × MultiF Seamless Assembly Mix from ABclonal (Wuhan, China) were added into 1.5 mL tube and incubated at 50 °C for 15 min. Then, the mixture was transformed into 100 μL competent cells (*E. coli* DH5α) using heat shock and plated on LB agar with appropriate antibiotics. The results was confirmed by sequencing. The constructed plasmids were then used for protein expression and whole-cell biotransformation. Especially, the fragments of *Tp*CAR and *Bs*Sfp were ligated to the pRSFDuet−1 vector, generating pRSFDuet−1-T7-RBS-6His-*Tp*CAR-RBS-*Bs*Sfp. The engineered strain IAA was obtained according to the previously reported protocols[32,38]. Briefly, the sequences of sgRNA for different target genes (*dkgB*, accession number: B21_00205; *yeaE*, accession number: B21_01738; *yahK*, accession number: B21_00284; *yjgB*, accession number: B21_04099 and *yqhC*, accession number: B21_02833) were obtained from online software (https://www.atum.bio/eCommerce/

cas9/input) and ligated to the plasmid pTargetF (Addgene), giving the modified pTargetF. The upstream and downstream homologous arms of target genes were amplified and merged, resulting in deletion fragments. The plasmid pEcCas (Addgene) was first transformed into *E. coli* BL21 (*DE3*) by chemical transformation and then used for the preparation of the electroporated competent cells. Next, the corresponding deletion fragment and the modified pTargetF were co-transformed into the above competent cells and then plated onto LB agar with appropriate antibiotics. The correct single colony was selected and confirmed by sequencing. The plasmids pTargetF and pEcCas were further removed by the corresponding culture conditions with appropriate antibiotics and sugar, finally giving the engineered strain IAA. The plasmids for protein expression were transformed into *E. coli* BL21 (*DE3*), while the plasmids for the synthesis of PEIAs were transformed into the engineered strain IAA.

### Expression and purification of recombinant proteins
A single colony of recombinant *E. coli* BL21 (*DE3*) containing *Tf*NCS was inoculated into LB medium supplemented with ampicillin (100 mg L$^{-1}$) and cultured overnight at 37 °C. Then, overnight cultures were inoculated to 2YT medium containing ampicillin (100 mg L$^{-1}$). Protein expression was induced with 0.2 mM β-D-1-thiogalactopyranoside at 18 °C for 12 h when the optical density (OD$_{600}$) of engineered strain reached 0.6 – 0.8. Next, the cells were collected by centrifugation at 6000 *g* for 10 min at 4 °C. The above pellets were resuspended with lysis buffer (25 mM Tris-HCl pH 8.0, 300 mM NaCl, 10 mM imidazole, and 5% (w/v) glycerol) at a concentration of 0.1 g mL$^{-1}$. The resuspended solution was lysed by the high-pressure homogenizer (Union-Biotech Co., Ltd., Shanghai, China), followed by centrifugation at 30,000 *g* for 30 min at 4 °C. The above supernatant was collected and used for target protein purification by NI-NTA agarose column. The collected protein was desalted to remove imidazole through the desalting column (buffer: 25 mM Tris-HCl pH 8.0 and 150 mM NaCl). The purified protein was stocked at −80 °C for further study. Other proteins were also purified by using the same method. The purified *Tp*CAR was named as holo-*Tp*CAR as it was activated by *Bs*Sfp in vivo.

### Enzymatic activity assay
To determine the catalytic activity of *Gs*OMT1 and *Rn*COMT towards (*S*)−**1b**, 5 μM *Gs*OMT1 or 5 μM *Rn*COMT, 10 mM MgCl$_2$, 2 mM (*S*)−**1b**, and 4 mM SAM (pH 7.5) were incubated in a 200 μL reaction system containing 0.1 M HEPES pH 7.5 and 0.1 M NaCl for 2 h at 30 °C. Then, the above reaction was quenched by adding a 4-fold volume of acetonitrile and centrifuged for 5 min at 20000 *g* to prepare samples. The samples were analyzed by a HPLC system (Waters 2695) with a 2996 photodiode array (PDA) detector equipped with a C18 reverse-phase column (4.6 × 250 mm, 5 μm) at 30 °C, and detected at a 280 nm wavelength with a flow rate of 1 mL·min$^{-1}$. A linear gradient elution method A (0–1 min, 5% solvent A; 1–21 min, 60% solvent A; 21–21.5 min, 95% solvent A; 21.5–24.5 min 95% solvent A; 24.5–25 min 5% solvent A; 25–30 min 5% solvent A) was used. Solvent A contains acetonitrile with 0.1% trifluoroacetic acid (TFA) and solvent B contains water with 0.1% trifluoroacetic acid (TFA). The catalytic activity of CNMT and *Gs*OMT1 was assessed by detecting the titer of (*S*)−**1d** and (*S*)−**12c**, respectively.

To evaluate the catalytic activity of NMT enzymes (*Gs*NMTt, PavNMT, TNMT and CNMT), a cascade reaction containing *Rn*COMT and NMTs was constructed. 5 μM *Rn*COMT, 25 μM NMTs, 10 mM MgCl$_2$, 2 mM (*S*)−**1b** and 8 mM SAM (pH 7.5) were incubated in a 200 μL reaction system containing 0.1 M HEPES pH 7.5 and 0.1 M NaCl for 4 h at 30 °C. Then the reaction was quenched by adding a 4-fold volume of acetonitrile. Samples were prepared and analyzed using the same method as described above.

For the catalytic activity of *Gs*OMT1 or its mutants towards (*S*)−**12b**, 5 μM *Gs*OMT1, 1.60 mM (*S*)−**12b**, and 4 mM SAM (pH 7.5) were incubated in a 200 μL reaction system containing 0.1 M HEPES pH 7.5 and 0.1 M NaCl for 2 h at 30 °C. Then, the reaction was quenched by adding 200 μL acetonitrile and 1% TFA. Samples were prepared and analyzed by the same method as described above.

### Optimum pH and temperature of *Gs*OMT1
To determine the optimal pH and temperature for the *O*-methylation of (*S*)−**12b** catalyzed by *Gs*OMT1, the conversion of (*S*)−**12b** to (*S*)−**12c** was utilized. The experimental procedure and method employed for sample analysis follows a similar approach to the catalytic activity assay. Potassium phosphate buffer (pH 5.5–7.5, 200 mM), Tris-HCl buffer (pH 7.5–9.0, 200 mM) and Glycine-NaOH (pH 9.0–10.0, 200 mM) were used to determine the optimum pH. The optimal reaction temperature was determined by a range of temperatures from 20 °C to 45 °C with an interval of 5 °C.

### Kinetic assay of *Gs*OMT1
The kinetic parameters ($V_{max}$, $K_m$, $k_{cat}$) were determined by assessing the initial reaction rates at different substrate concentrations at 30 °C in 200 mM potassium phosphate buffer, pH 7.5. The experimental procedure and method used for sample analysis were similar to the catalytic activity assay. The Michaelis-Menten equation was used to obtain the kinetic parameters.

### Molecule docking
The Glide module in Schrödinger 2021 was used for all docking analysis in this study. The structure of CNMT used in the docking analysis was obtained from Protein Data Bank (PDB ID: 6GKY, coclaurine *N*-Methyltransferase). All water molecules and substrate F2Z in the crystal structure were removed, and followed by adding hydrogen atoms to the protein. The docking site was defined based on the coordination of substrate F2Z from 6GKY. The docking radius was set to 9.0 Å according to the cavity volume of the substrate pocket. Then, (*S*)−**1c** was prepared by generating multi-conformations and was docked into the CNMT/SAH complex using the Glide module. By comparing energy score and geometric conformation, the appropriate complexes were selected for further analysis.

The structure of *Gs*OMT1 used in the docking analysis was predicted by Alphafold2. The *Gs*OMT1/SAM binary complex was generated by overlapping with the complex crystal of (*S*)-norcoclaurine 6-*O*-methyltransferase (PDB ID: 5ICE). The docking site was defined based on the coordination of substrate 2H4 from 5ICE. The docking radius was set to 9.5 Å according to the cavity volume of the substrate pocket. Then, (*S*)−**12b** was prepared by generating multi-conformations and docked into the *Gs*OMT1/SAM complex using the Glide module. By comparing energy score and geometric conformation, the appropriate complexes were selected for further analysis.

### In vitro reactions
To determine the feasibility of multi-enzyme cascade (in vitro), the one-pot two-step strategy was applied in vitro. In the PEIA moiety module, the reactions were performed in the buffer containing 0.1 M HEPES pH 7.5 and 0.1 M NaCl at 30 °C for 4 h. 5 μM *holo-Tp*CAR, 5 μM *Tf*NCS, 5 μM GDH, 10 mM MgCl$_2$, 4 mM ATP (pH 7.5), 1 mM NADP$^+$, 10 mM glucose, 2 mM sodium ascorbate, 2 mM dopamine, and 2 mM substrate **1** were added to the above system to initiate the reaction and a final volume of reaction system was 200 μL. After 4 h, the above reaction mixture was boiled at 95 °C for 5 min, and then centrifuged at 20000 *g* for 5 min to collect the supernatant. In the MT module, 5 μM *Rn*COMT, 25 μM CNMT* and 8 mM SAM (pH 7.5) were directly added to the collected supernatant to initiate *O*-methylation and *N*-methylation of (*S*)−**1b** at 30 °C for 4 h. The above reaction was quenched by adding a fourfold volume of acetonitrile. All samples were prepared by centrifugation for 5 min at 20000 *g* and then filtered with 0.22 μm filter before HPLC analysis. The HPLC analysis method is the same as the linear gradient elution method A.

## Whole-cell biocatalysis (in vivo)

The cells used for the whole-cell reactions were harvested, and washed by HEPES buffer. For the nine strains (M1A-M1I) containing different combinations of plasmids in the PEIA moiety module, the reactions were conducted in a 50 mL falcon tube with a final volume of 5 mL. Fresh cells were resuspended in the buffer containing 0.05 M HEPES pH 7.5 and 0.2 M KCl, reaching a final $OD_{600}$ of 6. 20 mM $MgCl_2$, 20 mM glucose, 5% DMSO, 7.5 mM substrate **1**, 5 mM dopamine and 5 mM sodium ascorbate were added to the above system to initiate the reaction at 30 °C and 250 rpm for 8 h. After 8 h, 200 µL samples were taken and quenched by a fourfold volume of acetonitrile. The samples were prepared by centrifugation for 5 min at 20,000 $g$, and then filtered with 0.22 µm filter before HPLC analysis. The HPLC analysis method was the same as the method A.

One-pot two-step: in the PEIA moiety module, the fresh cells of M1G strain were resuspended in the buffer containing 0.05 M HEPES pH 7.5 and 0.2 M KCl, reaching a final $OD_{600}$ of 6. 20 mM $MgCl_2$, 20 mM glucose, 5% DMSO, 7.5 mM substrate **1**, 5 mM dopamine and 5 mM sodium ascorbate were added to the above system to initiate the reaction at 30 °C and 250 rpm. After 8 h, the reaction supernatant was collected by centrifugation at 6000 $g$ for 10 min, and then used to resuspend the fresh cells of strains S1-S6. 30 mM L-Met, and 5 mM ATP were added to initiate the reaction at 30 °C and 250 rpm. After 8 h, The above reaction was quenched by adding a 4-fold volume of acetonitrile. The sample were prepared by centrifugation for 5 min at 20,000 $g$ and then filtered with 0.22 µm filter before HPLC analysis. The HPLC analysis method was the same as the method A.

One-pot three-step: in the PEIA module, the process was the same as the section of "One-pot two-step". After 8 h, the reaction supernatant of the strain M1G was collected by centrifugation at 6000 $g$ for 10 min, and then used to resuspend the fresh cells of the strain M2B (*Rn*COMT), reaching a final $OD_{600}$ of 9. 30 mM L-Met and 5 mM ATP were added to initiate the reaction at 30 °C and 250 rpm. After 6 h, the reaction supernatant of the strain M2B (*Rn*COMT) was collected by centrifugation at 6000 $g$ for 10 min, and then was adjusted to pH 7.5 using 5 M NaOH. The supernatant was used to resuspend the fresh cells of the strain M3B, reaching a final $OD_{600}$ of 9. 5 mM ATP was added to initiate the reaction at 30 °C and 250 rpm. After 6 h, 200 µL samples were taken and quenched by a 4-fold volume of acetonitrile. The samples were prepared by centrifugation at 20,000 $g$ for 5 min, and filtered with 0.22 µm filter before HPLC analysis. The HPLC analysis method was the same as the method A.

In order to efficiently synthesize (*S*)−**12c**, M2B (sumo-*Gs*OMT1$^{V125Q}$) was constructed. The fresh cells of M2B (sumo-*Gs*OMT1$^{V125Q}$) were resuspended using the reaction supernatant of the PEIA moiety module, reaching a final $OD_{600}$ of 42. The preparation and analysis procedures for (*S*)−**12d** were consistent with the previous description. The titer and yield of (*S*)−**1d**−(*S*)−**12d** were determined by HPLC analysis against product standards at 280 nm.

## Preparation of racemic 1b and chiral HPLC analysis of (*S*)−1b

Similar to the whole-cell reaction section, HEPES buffer (pH 7.5) was replaced by KPI buffer (pH 7.5) to prepare racemic **1b**. The stereoselectivity of (*S*)−**1b** was analyzed by a chiral HPLC system (Waters 2695) equipped with a 2996 photodiode array (PDA) detector and a CHIRALCEL OD-H column (4.6 × 250 mm, 5 µm) at 30 °C, and detected at a 280 nm wavelength with the flow rate of 0.5 mL min⁻¹. An isocratic elution method (hexane: 2-propanol: diethylamine = 72:28:0.1) was used based on the previous report[46].

## Substrate synthesis of 12

Chemical synthesis of (*E*)-3-hydroxy-4,5-dimethoxycinnamic acid was as follows: 3-hydroxy-4,5-dimethoxybenzaldehyde (0.50 g, 2.75 mmol), malonic acid (0.86 g, 8.25 mmol), and piperidine (0.1 mL) were refluxed in pyridine (10 mL) at 120 °C for 6 h. After that, 5 M HCl (30 mL) was used to quench the mixture and extracted by dichloromethane (3 × 30 mL). Then, the extraction solution was washed with 5 M HCl (30 mL) twice, and washed with 30 mL brine. The organic layer was dried with anhydrous sodium sulfate, and concentrated in vacuum. The sample was further purified by column chromatography (dichloromethane:methanol = 20:1, with 0.1% formic acid) to yield (*E*)-3-hydroxy-4,5-dimethoxycinnamic acid as a light yellow solid. A solution containing (*E*)-3-hydroxy-4,5-dimethoxycinnamic acid (0.40 g, 1.79 mmol) in 10% sodium hydroxide (3 mL) and ethanol (6 mL) was hydrogenated at room temperature and atmospheric pressure using 10% Pd/C (20 mg) as a catalyst until the uptake of hydrogen ceased. The catalyst was filtered, and the solution was concentrated under vacuum. Subsequently, it was acidified with HCl to pH 2.0 and then extracted with dichloromethane (3 × 10 mL). The organic layer was washed with 30 mL brine, dried with anhydrous sodium sulfate, and concentrated in vacuum to obtain a faint yellow solid[47].

## Preparation of final products

Similar to the whole-cell reaction section, a 100 mL reaction solution was used to prepare final products. As for (*S*)−**1d**, the final reaction solution was centrifuged to collect the supernatant. Subsequently, the pH of supernatant was adjusted to ten by adding 5 M NaOH, followed by extraction with dichloromethane (3 × 100 mL). The organic phase was dried with anhydrous sodium sulfate and concentrated under vacuum. The resulted sample was dissolved in methanol and further purified by semi-preparative HPLC (Welch, Ultimate XB-phenyl 10 × 250 mm, 5 µm). The preparation method for other final products, except for (*S*)−**12d**, was similar to that of (*S*)−**1d**. The semi-preparative HPLC conditions were shown in Supplementary Table 6. NMR data were collected on Bruker Avance spectrometers (400 or 600 MHz for ¹H NMR, and 101 or 151 MHz for ¹³C NMR). NMR spectra are presented in the Supplementary Figs. 19–33. HRMS data were collected on a Waters Maldi Synapt UPLC-MS system. LC-MS data are presented in the Supplementary Figs. 34–48.

## Preparation and characterization of intermediates

Similar to whole-cell reaction section, a 100 mL solution was used to prepare intermediates. For (*S*)−**1b**, the solution was centrifuged to collect the supernatant, which was subsequently removed by freeze-vacuum drying and then dissolved in methanol. The intermediate was purified by semi-preparative HPLC (Angilent, ZORBAX Eclipse XDB C18 column, 9.4 × 250 mm, 5 µm). The preparation method of (*S*)−**12b** was the same as (*S*)−**1b**. The preparation method for (*S*)−**12c** was identical to that of (*S*)−**1b**, except that the $OD_{600}$ of the strain M2B (sumo-*Gs*OMT1$^{V125Q}$) was 42. The semi-preparative HPLC conditions are shown in Supplementary Table 6.

## Scale-up preparation of (S)-12d

Similar to whole-cell reaction section, (*S*)−**12d** was scaled up to a 300 mL solution. After the reaction, the pH of supernatant was adjusted to 8.0 by adding saturated $NaHCO_3$ and subsequently extracted by dichloromethane (3 × 300 mL). The organic phase was dried with anhydrous sodium sulfate and concentrated under vacuum. The resulted sample was dissolved in methanol and purified by semi-preparative HPLC (Welch, Ultimate XB-phenyl 10 × 250 mm, 5 µm). The semi-preparative HPLC conditions are shown in Supplementary Table 6.

## Statistics and reproducibility

Sample size was not predetermined using any statistical method. All data were included in the analyses. The experiments were not randomized. The Investigators were not blinded to allocation during experiments and outcome assessment.

**Reporting summary**

Further information on research design is available in the Nature Portfolio Reporting Summary linked to this article.

## Data availability

All data supporting the findings of this study are provided within the paper and its Supporting Information files. Source data are provided within this paper. The accession codes of CNMT (6GKY), PavNMT (5KN4), TNMT (6P3M) and (S)-norcoclaurine 6-O-methyltransferase (5ICE) can be found in the Protein Data Bank (PDB) (https://www.rcsb.org) and are also indicated in Chemicals and Materials section. Source data are provided with this paper.

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

## Acknowledgements

This work was supported by the National Key R&D Program of China (2018YFA0901700 to Y.R.), the National Natural Science Foundation of China (32270082 to Y.R. and 22108122 to Z.L.), the Natural Science Foundation of Jiangsu Province (BK20202002 to Y.R.) and Postgraduate Research & Practice Innovation Program of Jiangsu Province (KYCX20_1814 to Y.G.).

## Author contributions

Y.R. supervised and designed the project; Y.G., F.L., Z.L., Z.D., Y.Z., Z.Y., and C.L. performed research and data analysis; Z.L. and Z.Y. contributed to data analysis; Y.G., Z.L., and Y.R. wrote the paper.

## Competing interests

The authors declare no competing interests.
