## [Peer Review File · Nature Communications]

REVIEWER COMMENTS

Reviewer #1 (Remarks to the Author):

In this manuscript, Gao et al set out to establish a biocatalytic cascade for the synthesis of phenethylisoquinoline alkaloid (PEIA) precursors and their non-natural derivatives in order to alleviate challenges in sourcing and synthesizing these compounds. To do this, they propose a 4-enzyme route to produce these compounds, then systematically identify enzymes that can carry out the necessary chemical transformations. The authors are successful in building a working biocatalytic cascade, and they refine and improve this strategy through a combination of rational enzyme mutagenesis and metabolic engineering. Finally, they demonstrate that their biocatalytic cascade can be used to build a variety of different PEIA end-products in high yields by using different starting chemicals.

This work is a nice example of rational design for the biocatalytic synthesis of plants alkaloids and their unnatural analogs. In general, the manuscript is well-written and easy to follow and rigorous in data presentation. While I think that the manuscript is strong, there are several items that would be useful to address prior to publication, as outlined below.

Major comments

- I have a few general comments on the solution(s) that this manuscript is seeking to provide. First, the work in this manuscript is introduced as important for the sourcing of plant alkaloids. While many alkaloids are challenging to source for a variety of reasons, including low abundance, this isn't universally true, and often, medicinal alkaloids accumulate to relatively high levels in the plant (this is true of colchicine, which can be isolated at high milligram per gram dry weight levels from the native plant). In my opinion, statements indicating that plant compounds are always present at low abundances should be altered for a more accurate and nuanced depiction (e.g. lines 16 and 29). Second, it is stated several times that achieving the biocatalytic synthesis of (S)-autumnaline will provide a platform for engineering the synthesis of colchicine. To my knowledge, (S)-autumnaline is not used as a synthetic precursor for the production of colchicine, except within the native biosynthetic pathway, which this paper seeks to circumvent (e.g. due to the requirement of more cytochrome P450s). So while I do think that the work performed herein is important, I'm not sure that it provides any new access to colchicine itself, and it is not clearly/specifically stated how these precursors could otherwise be utilized. If there is a precedent or rationale for this, then it would be helpful to provide an example or citation. Otherwise, I would consider reframing.
- Line 47 – Could you please define what is meant by multi-enzyme cascade? It seems ambiguous about whether this is in vivo (i.e. metabolic engineering) or in vitro. Similarly, it would be helpful to explicitly define things like 'one-pot, multi-step' and 'one-pot two-step' throughout the manuscript, as it isn't

always immediately clear what is being performed/compared in each assay. Likewise, please be clear about how each biocatalysis reaction setup is being performed (e.g. when/how are compounds being isolated and fed to other reactions or *E. coli* strains).

Minor comments

- Line 13 – plant alkaloids have been used as pharmaceuticals for a long time (hundreds of years, or more), so it isn't totally accurate to say that they are 'emerging'.
- Line 20 – I would recommend removing 'impressive'
- Line 56 – "These appealing advantages of multistep biocatalytic cascades have revolutionized the manufacture of natural or unnatural products." Can a citation/example be given for this? It isn't immediately clear to me that it has revolutionized how we source these compounds.
- Line 74 – define 'PP aldehydes' in the text
- Could you please define what is being measured when you report 'Relative activity' throughout the manuscript? (is this turnover?)
- Line 97 – specify about what is meant by the 'most complex compound' - the most complex methoxylated compound?
- Line 100 – To my knowledge, it isn't totally accurate to refer to the 'NCS from *G. superba*'. NCS specifically refers to noroclaurine synthase from benzyloisoquinoline alkaloid biosynthesis. Colchicine biosynthesis is predicted to proceed via a similar Pictet-Spengler condensation reaction (and NCS was used to engineer this), but it is not yet known what type of enzyme catalyzes this reaction in *G. superba* (and it is likely not an homolog of NCS).
- Line 120 and 122 – Though your approach of rational enzyme mutagenesis is successful, I don't think it is accurate to call your approach 'directed evolution' (from your description, there are no rounds of evolution/selection).
- Line 129 – Could you provide some detail on how this semi-rational design was performed?
- Line 146 – In the introduction, the authors introduce how reconstitution of biosynthetic pathways in microbes can be problematic, thus prompting the biocatalytic cascade approach. However, in this section, the authors take the opposite stance in favor of metabolic engineering in *E. coli*. I don't think any major changes are necessary, but it would be useful to be clear throughout the paper about the intent of this work (e.g. is the goal in the end to have strains for metabolic engineering, to avoid this entirely for in vitro synthesis, or some combination of the two?)
- Line 152 – It is a little confusing that the RARE strain had already been made; it seems that the same thing is being done here but starting with a different strain of *E. coli*? (is it appropriate to give it the same name?).
- Line 156 – if knockout of the genes is really crucial, then why doesn't M1A work better than WT?
- Line 159 – I would recommend removing 'great'

- Fig 3 – Can you define what the predicted ‘by-product’ is?
- Line 182 – From Fig 4b, it isn’t immediately clear that plasmid copy number is the variable being tested. Could you specify in the methods or figure caption what the relative plasmid copy number for each construct is for reference?
- Line 190 (and Fig 4e) – Is the data for (S)-1c missing from this figure?
- Line 195 – much like described above; could you add details about how this experiment is distinct from the ‘one-pot, two-step’ method?
- Line 239 –GsOMT1 is already introduced earlier in the manuscript, so it feels somewhat strange to re-introduce it here as though it is a different enzyme than previously analyzed.
- Line 246 – Can additional info be provided on how the mutants were designed? (i.e. how did the authors pick the specific mutations to analyze)
- Fig 6c – could you include the specific modules used here within the figure caption?
- Line 272 – “After optimizing the concentration of M2B (sumo-GsOMT1V125Q) (Supplementary Fig. 18), 6 mM 12 and 4 mM dopamine could be converted to 2.14 mM (S)-autumnaline with a moderate yield of 53.50% (Fig. 6c), which is 2.25 times higher than that of the cascade reaction using M1G, M2B and M3B as catalysts.” -> I would consider rephrasing this sentence for clarity

Reviewer #2 (Remarks to the Author):

The authors present an interesting biochemical scheme for production of new molecules. There are a few aspects that were not well described in the paper:

Availability/synthesis of the PP-acid was not well-described. How will this be conducted (via biological or chemical means)?

The overall promiscuity and evolution of the NCS enzyme in the 1st module should be discussed further for varying PP-acids. How promiscuous is this system? How much re-engineering will need to be done to produce other PEIAs? How will this selection take place?

The authors indicate that error bars are duplications, but it is not clear what these truly represent (biological or technical)?

Where is the remainder of the carbon going in these timecourses? It would be helpful to see other side products as the yields are not pure bioconversion yields here.

Reviewer #3 (Remarks to the Author):

The author demonstrated a *E. coli*-based bioproduction of phenethylisoquinolines fed with pp-acids and dopamine. *E. coli* has natural pathway to convert pp-acids to the corresponding aldehyde. The strategies of using a shunt pathway & structural diversity of fed substrates have been previously utilized and demonstrated in the production of benzyloisoquinoline alkaloids in *E. coli*. To enhance the conversion towards the final products by the unnatural substrates, the team has conducted protein engineering to make the biocatalyst of each step more efficient. The authors have also engineered *E. coli* to decrease the consumption of the substrate. Overall, they were able to achieve a significant increase in conversion and a very encouraging titer (~ g/L) for all the natural and unnatural phenethylisoquinolines.

There are several major drawbacks that the author needs to address in the discussion: 1) it's true that the author was able to detour from the CYPs using fed substrates, but down the pathway towards colchicine for example, there are many more CYPs; 2) how much is the fed substrate and a simple economic analysis will be helpful for the audience to appreciate the strategy more; 3) the high conversion is achieved through a multiple stage whole cell biotransformation, rather than single stage fermentation, which could be time and effort consuming - a discussion on the con and pro of the established process would be helpful for the audience.

Also, please make sure to proof read the figures and the manuscript, for example, the red line should be 12d instead of 12c in fig 6c.

Response to reviewers:

Reviewer #1 (Remarks to the Author):

In this manuscript, Gao et al set out to establish a biocatalytic cascade for the synthesis of phenethylisoquinoline alkaloid (PEIA) precursors and their non-natural derivatives in order to alleviate challenges in sourcing and synthesizing these compounds. To do this, they propose a 4-enzyme route to produce these compounds, then systematically identify enzymes that can carry out the necessary chemical transformations. The authors are successful in building a working biocatalytic cascade, and they refine and improve this strategy through a combination of rational enzyme mutagenesis and metabolic engineering. Finally, they demonstrate that their biocatalytic cascade can be used to build a variety of different PEIA end-products in high yields by using different starting chemicals.

This work is a nice example of rational design for the biocatalytic synthesis of plants alkaloids and their unnatural analogs. In general, the manuscript is well-written and easy to follow and rigorous in data presentation. While I think that the manuscript is strong, there are several items that would be useful to address prior to publication, as outlined below.

A. We appreciate your positive evaluations and valuable suggestions, which help us improve the quality of this manuscript. Here we prepared this point-to-point response and highlighted the changes in blue.

Major comments

Q1: I have a few general comments on the solution(s) that this manuscript is seeking to provide. First, the work in this manuscript is introduced as important for the sourcing of plant alkaloids. While many alkaloids are challenging to source for a variety of reasons, including low abundance, this isn't universally true, and often, medicinal alkaloids accumulate to relatively high levels in the plant (this is true of colchicine, which can be isolated at high milligram per gram dry weight levels from the native plant). In my opinion, statements indicating that plant compounds are always present at low abundances should be altered for a more accurate and nuanced depiction (e.g. lines 16 and 29).

A1: Thanks for the reviewer's objective and professional comment. According to your suggestion, we have deleted this phrase "their low contents in plants" in the **Abstract** section. Meanwhile, we changed the sentence "However, their content in medicinal plants is always low. Moreover, they are also susceptible to weather, climate change, pest and locations, risking their supply." as follows in the manuscript (lines 28-30):

"However, their content in medicinal plants are susceptible to weather, climate change, pest and locations, risking their market supply."

Q2: Second, it is stated several times that achieving the biocatalytic synthesis of (S)-autumnaline will provide a platform for engineering the synthesis of colchicine. To my knowledge, (S)-autumnaline is not used as a synthetic precursor for the production of colchicine, except within the native biosynthetic pathway, which this paper seeks to circumvent (e.g. due to the requirement of more cytochrome P450s). So while I do think that the work performed herein is important, I'm not sure that it provides any new access to colchicine itself, and it is not clearly/specifically stated how these precursors could otherwise be utilized. If there is a precedent or rationale for this, then it would be helpful to provide an example or citation. Otherwise, I would consider reframing.

A2: Thanks for your question. The artificial biosynthetic pathway constructed in this work could obtain the precursor of colchicine (S)-autumnaline through an efficient and short synthetic pathway without the employment of P450 CYP75A109. Meanwhile, the precursors could be fed to microorganisms for the biosynthesis of final products. For instance, magnoflorine and scoulerine also as the isoquinoline alkaloids, were biosynthesized by feeding the precursor (S)-reticuline in *S.cerevisiae* (Minami, H. et al. *Proc. Natl. Acad. Sci. U.S.A.* **105**, 7393-7398 (2008)). Therefore, we believe that the artificial pathway constructed in this study efficiently realizes the synthesis of (S)-autumnaline (the precursor of colchicine), which will provide a platform for the biosynthesis of colchicine and other complex PEIAs.

Q3: Line 47 – Could you please define what is meant by multi-enzyme cascade? It seems ambiguous about whether this is in vivo (i.e. metabolic engineering) or in vitro.

A3: Thanks for your constructive comment. According to the reference (France, S.P. et al. *ACS Catal.* **7**, 710-724 (2017)), we have defined “multi-enzyme cascade” as “an enzymatic procedure involving two or more steps for producing valuable chemical compounds” in the manuscript (lines 47-50) as follows:

“To overcome these limitations, the multi-enzyme cascade reaction, which is an enzymatic procedure involving two or more steps for producing valuable chemical compounds from readily available (commercially available/easily accessible) precursor, maybe a powerful tool to sustainably synthesize valuable natural or unnatural products.”

Q4: Similarly, it would be helpful to explicitly define things like ‘one-pot, multi-step’ and ‘one-pot two-step’ throughout the manuscript, as it isn’t always immediately clear what is being performed/compared in each assay.

A4: Thanks for your great suggestion. The “one-pot, multi-step strategy” contains the strategy of “one-pot two-step or one-pot three-step”. We have defined “one-pot two-step *in vivo*” as “the strains containing the PEIA moiety module and the MT module are added consecutively” in the revised manuscript (lines 159-162) as follows:

“Firstly, we separately introduced the PEIA moiety module and the MT module into *E. coli* BL21 (*DE3*) to produce PEIAs in a one-pot two-step process, where the strains containing the PEIA moiety module and the MT module are added consecutively (Fig. 2a, 2b).”

In addition, we have also defined “one-pot two-step *in vitro*” as “the steps catalyzed by the PEIA moiety module and by the OMT module are performed in sequential mode” in the revised manuscript (lines 148-151) as follows:

“With the availability of CNMT*, we next evaluated the feasibility of the whole cascade reaction via a one-pot two-step process *in vitro*, in which the steps catalyzed by the PEIA moiety module and by the OMT module are performed in sequential mode, obviating the need for purification of the intermediates and avoiding the side reaction caused by

RnCOMT towards dopamine.”

Q5: Likewise, please be clear about how each biocatalysis reaction setup is being performed (e.g. when/how are compounds being isolated and fed to other reactions or E. coli strains).

A5: Thanks for your valuable suggestion. In order to facilitate readers' understanding, we have described each biocatalysis reaction setup in the related figure legend and the section of “**Materials and Methods**” as follows:

We have described the multi-enzyme cascade *in vitro* in a “one-pot two-step” process in the figure legend of Fig. 2 and the section of “**Materials and Methods**”. For the figure legend of Fig. 2 (lines 97-100) as follows: “In the PEIA moiety module, substrate **1**, dopamine, enzymes and cofactors were added to the reaction vessel and reacted for 4 h. The reaction solution was boiled and centrifuged to collect the supernatant. In the MT module, enzymes and co-factors were added to the resulting supernatant and incubated for 4 h.” For the section of “**Materials and Methods**”, please see lines 409-415.

We have also described whole-cell biocatalysis (*in vivo*) in a “one-pot two-step” process in the figure legend of Fig. 4 and the section of “**Materials and Methods**”. For the figure legend of Fig. 4 (lines 221-223) as follows: “Substrate **1** and dopamine were added to the suspension of the strain M1G and reacted for 8 h. M1G supernatant, obtained by centrifugation, was used to resuspend strains S1-S6 and reacted for 6 h.” For the section of “**Materials and Methods**”, please see lines 426-431.

In addition, we have described whole-cell biocatalysis (*in vivo*) in a “one-pot three-step” process in the figure legend of Fig. 4 and the section of “**Materials and Methods**”. For the figure legend of Fig. 4 (lines 228-231) as follows: “Substrate **1** and dopamine were added to the suspension of the strain M1G and reacted for 8 h. M1G supernatant, obtained by centrifugation, was used to resuspend strains M2A-M2C and reacted for 6 h. The resulting supernatant, adjusted to pH 7.5, was used to resuspend strains M3A-M3C and reacted for 6 h.” For the section of “**Materials and Methods**”, please see lines 433-439.

Minor comments

Q6: Line 13 – plant alkaloids have been used as pharmaceuticals for a long time (hundreds of years, or more), so it isn't totally accurate to say that they are 'emerging'.

A6: Thanks for your professional advice. We have changed the word “emerging” to “important”.

Q7: Line 20 – I would recommend removing 'impressive'

A7: Thanks for your great suggestion. We have changed the word “impressive” to “high”.

Q8: Line 56 – “These appealing advantages of multistep biocatalytic cascades have revolutionized the manufacture of natural or unnatural products.” Can a citation/example be given for this? It isn't immediately clear to me that it has revolutionized how we source these compounds.

A8: Thanks for your nice advice. According to your suggestion, we have cited two meaningful references for this sentence (Cai, T. et al. *Science* **373**, 1523-1527 (2021), Huffman, M.A. et al. *Science* **366**, 1255-1259 (2019)), which have been listed as ref. 25 and 29 (lines 58-59) in the revised manuscript as follows:

“These appealing advantages of multistep biocatalytic cascades have revolutionized the manufacture of natural or unnatural products^{25,29}.”

Q9: Line 74 – define 'PP aldehydes' in the text

A9: Thanks for your suggestion. We have defined the “PP aldehydes” as “phenylpropionic aldehydes” in the revised manuscript (74-75) as follows:

“It suggests that they could be produced from the PEIA moiety derived from phenylpropionic aldehydes (PP aldehydes) and dopamine, followed by methylation (Fig. 1c, 2a, 2b).”

In addition, we also defined “PP acids” as “phenylpropionic acids” in the manuscript

(lines 78-80) as follows:

“Considering that PP aldehydes are easily converted to alcohols in microorganisms, a carboxylic acid reductase (CAR) was introduced to convert phenylpropionic acids (PP acids) to the desired aldehydes *in situ*.”

Q10: Could you please define what is being measured when you report ‘Relative activity’ throughout the manuscript? (is this turnover?)

A10: Thanks for your professional advice. We have defined the “Relative activity” as “The titer of (S)-**1d** is defined as the catalytic activity of CNMT^{WT} and its mutants towards (S)-**1c**.” (Please see lines 93-94.), and Please see Lines 282-283: “The titer of (S)-**12c** is defined as the catalytic activity of GsOMT1^{WT} and its mutants towards (S)-**12b**.”

Q11: Line 97 – specify about what is meant by the ‘most complex compound’ - the most complex methoxylated compound?

A11: We apologize for inappropriate depiction. We have changed “the most complex compound” to “the compound” (lines 102-103).

Q12: Line 100 – To my knowledge, it isn’t totally accurate to refer to the ‘NCS from *G. superba*’. NCS specifically refers to norcoclaurine synthase from benzyloquinoline alkaloid biosynthesis. Colchicine biosynthesis is predicted to proceed via a similar Pictet-Spengler condensation reaction (and NCS was used to engineer this), but it is not yet known what type of enzyme catalyzes this reaction in *G. superba* (and it is likely not an homolog of NCS).

A12: Thanks for your professional suggestion. We have changed the sentence (lines 103-106) to “At first, NCSs from *Thalictrum flavum* (TfNCS) and *Coptis japonica* (CjNCS) were selected to determine their feasibility to synthesize the PEIA moiety since the related enzyme responsible for Pictet-Spengler condensation reaction from *G. superba* is not reported.”

Q13: Line 120 and 122 – Though your approach of rational enzyme mutagenesis is successful, I don't think it is accurate to call your approach 'directed evolution' (from your description, there are no rounds of evolution/selection).

A13: Thanks for your correction. We have changed all "directed evolution" to "structure-guided engineering" in the revised manuscript (lines 125-127 and line 128) as follows:

Please see lines 125-127: "To our delight, CNMT showed a very weak catalytic activity towards (S)-**1c** (Fig. 2b), suggesting that structure-guided engineering of CNMT is necessary to improve its catalytic activity."

Please see line 128: "Structure-guided engineering of CNMT to improve its catalytic activity towards PEIA."

Q14: Line 129 – Could you provide some detail on how this semi-rational design was performed?

A14: Thanks for your wonderful suggestion. We have provided the details on semi-rational design in this sentence (lines 135-139) as follows:

"Based on the results of alanine scanning mutagenesis, semi-rational design was conducted to further enhance the catalytic activity of CNMT. To establish potential hydrogen bonds between CNMT and the substrate (S)-**1c**, both L88 and F332 were mutated to serine. In addition, to obtain a better substrate binding conformation for substrate (S)-**1c**, N92 was mutated to valine. Finally, to investigate the steric effects of CNMT on the substrate (S)-**1c**, L88 was mutated to glycine and valine, and F332 was mutated to valine."

Q15: Line 146 – In the introduction, the authors introduce how reconstitution of biosynthetic pathways in microbes can be problematic, thus prompting the biocatalytic cascade approach. However, in this section, the authors take the opposite stance in favor of metabolic engineering in *E. coli*. I don't think any major changes are necessary, but it would

be useful to be clear throughout the paper about the intent of this work (e.g. is the goal in the end to have strains for metabolic engineering, to avoid this entirely for in vitro synthesis, or some combination of the two?)

A15: Thanks for professional suggestion. The intent of this work is a combination of multi-enzyme cascade and metabolic engineering. Although metabolic engineering has made progress in the synthesis of plant benzyloisoquinoline alkaloids, efficient synthesis of PEIAs in microorganisms by heterologous reconstruction is still challenging because of their intricate biosynthetic pathway and the difficulty in expressing plant-derived P450 enzymes. Therefore, in order to solve the above problems, combining metabolic engineering with multi-enzyme cascade may be a good strategy to solve the problems faced in constructing cell factories for efficient synthesis of plant natural products.

According to your suggestion, we have indicated the intention of this work in the revised manuscript (lines 53-55, lines 65-67 and lines 327-330) as follows:

Please see lines 53-55: "Furthermore, combined with the strategy of one-pot multi-step and metabolic engineering, which will circumvent the issues of the burden of protein expression, deficiency of cofactor and side reactions in a single cell, the titer of target products could be improved."

Please see lines 65-67: "After optimizing the process of four enzymes through metabolic engineering, various (S)-autumnaline derivatives are efficiently produced."

Please see lines 328-331: "To further increase the yields of final products, the strategies of enzyme engineering and metabolic engineering could be combined to optimize multi-enzyme cascade, and then effectively address the shortcomings of multi-stage biotransformation, such as several centrifugation and resuspension steps and ATP regeneration."

Q16: Line 152 – It is a little confusing that the RARE strain had already been made; it seems that the same thing is being done here but starting with a different strain of E. coli? (is it appropriate to give it the same name?).

A16: Thanks for your professional suggestion, we have changed all “RARE” to “IAA (improvement of aldehyde accumulation)” in the revised manuscript (line 165) and Supporting information.

Q17: Line 156 – if knockout of the genes is really crucial, then why doesn't M1A work better than WT?

A17: Thanks for your question. Although the two strains WT and M1A express the same proteins, the plasmid vectors carrying the target protein in the two strains are different, which can affect the expression level of the target protein and thereby influence product biosynthesis (Zhou, Y. et al. *Angew. Chem. Int. Ed.* **55**, 11647-50 (2016), Wang, F et al. *Nat. Commun.* **11**, 5035 (2020)). Therefore, it is unsuitable to assess the importance of knocked-out genes by comparing the titer of (S)-**1b** produced by strains WT and M1A.

To demonstrate the significance of the knocked-out genes in enhancing the titer of (S)-**1b**, it might be more appropriate to compare the titer of (S)-**1b** produced by strains WT and M1G. The key distinction between the two strains is that M1G is derived from WT by knocking out seven genes (*dkgB*, *yeaE*, *dkgA*, *yqhD*, *yahK*, *yjgB* and *yqhC*). The results indicated that the genes knocked-out indeed increased the titer of (S)-**1b** by 3.57 times.

Q18: Line 159 – I would recommend removing ‘great’

A18: Thanks for your advice. We have removed the word “great”.

Q19: Fig 3 – Can you define what the predicted ‘by-product’ is?

A19: Thanks for your professional advice. We have defined the ‘by-product’ as “3-(3,4,5-trimethoxyphenyl) propan-1-ol (**1a'**)” in the manuscript (lines 168-170) and Fig. 3 as follows:

Please see lines 168-170: “This finding well supports the notion that knock out of the above seven genes indeed inhibits the production of by-product 3-(3,4,5-trimethoxyphenyl) propan-1-ol (**1a'**), thereby increasing the titer of (S)-**1b**.”

Meanwhile, the byproduct has also been numbered as **1a'** in Fig 3 as follows:

Fig. 3. Biosynthesis of (S)-1b in engineered strain IAA.

Q20: Line 182 – From Fig 4b, it isn't immediately clear that plasmid copy number is the variable being tested. Could you specify in the methods or figure caption what the relative plasmid copy number for each construct is for reference?

A20: We apologize for ambiguous description of Fig. 4b. According to your constructive suggestion, we have updated the Fig. 4b as follows.

Fig. 4. Biosynthesis of (S)-1d in engineered strain IAA.

In addition, the plasmid copy number was added to figure legend of Fig. 4 and Supplementary Table 3. Please see lines 226-227: “The copy number of pRSFDuet, pETDuet and pCDFDuet are >100, ~40 and 20-40 respectively.”

Q21: Line 190 (and Fig 4e) – Is the data for (S)-1c missing from this figure?

A21: Thanks for your careful and professional review to raise this issue. According to previous report (Law, B.J.C. et al. *Angew. Chem. Int. Ed.* **55**, 2683-2687 (2016), Subrizi, F. et al. *Angew. Chem. Int. Ed.* **60**, 18673-18679 (2021)), due to the poor regioselectivity of *RnCOMT*, the major product is (S)-1c when *RnCOMT* catalyzes the substrate (S)-1b, with

a minor formation of the 7-methoxy isomer of (S)-**1c**. Furthermore, the separation and purification of (S)-**1c** and its 7-methoxy isomer are extremely challenging, resulting in the inability to quantify (S)-**1c**. Therefore, in Fig. 4e, we only monitored the time course of substrate (S)-**1b** and the final product (S)-**1d**.

Meanwhile, we apologize that it is not rigorous to present the data of (S)-**1c** in Fig. 4b, so the titer of (S)-**1c** is deleted in Fig. 4b.

Q22: Line 195 – much like described above; could you add details about how this experiment is distinct from the ‘one-pot, two-step’ method?

A22: We apologize for the unclear description. We have clarified the process of “one-pot two-step *in vivo*” and “one-pot three-step *in vivo*” in the above answer 5 (**A5**). Moreover, the experimental process was described in the figure caption of Fig. 4 and the section of “**Materials and Methods**” in the revised manuscript.

For the experimental process of “one-pot two-step *in vivo*”, please see lines 221-223 (Figure legend of Fig. 4) and lines 426-431 (**Materials and Methods**).

For the experimental process of “one-pot three-step *in vivo*”, please see lines 228-231 (Figure legend of Fig. 4) and lines 433-439 (**Materials and Methods**).

Q23: Line 239 –GsOMT1 is already introduced earlier in the manuscript, so it feels somewhat strange to re-introduce it here as though it is a different enzyme than previously analyzed.

A23: Thanks for your nice advice. We have changed the sentence to “Then, original OMT from *G. superba* (GsOMT1) was selected.” in the revised manuscript. Please see lines 257-258.

Q24: Line 246 – Can additional info be provided on how the mutants were designed? (i.e. how did the authors pick the specific mutations to analyze)

A24: Thanks for your valuable suggestion. We have provided the description on how to select the specific mutations in the revised manuscript (lines 264-268) as follows:

“In order to enhance the binding affinity between the substrate (**(S)-12b**) and GsOMT1, a potential hydrogen bonding network was constructed by mutating I122 and V125 to polar or charged amino acids. Meanwhile, to strengthen the hydrophobic interaction between the substrate (**(S)-12b**) and GsOMT1, E310 was mutated to nonpolar amino acids. Furthermore, to validate the hydrogen bonding interaction between the substrate (**(S)-12b**) and N181, N181 was mutated to alanine and glutamine.”

Q25: Fig 6c – could you include the specific modules used here within the figure caption?

A25: Thanks for your insightful suggestion. We have added the reaction process of substrate **12** and dopamine to produce (**(S)-12d**) in Fig. 6c as follows:

Fig. 6. Rational design of GsOMT1 for efficient biosynthesis of (S)-autumnaline.

Moreover, we have described the process of producing (S)-**12d** in the figure caption of Fig. 6 in the revised manuscript (lines 285-288) as follows:

“Substrate **12** and dopamine were added to the suspension of the strain M1G and reacted for 8 h. M1G supernatant, obtained by centrifugation, was used to resuspend the strain M2B (sumo-GsOMT1^{V125Q}) and reacted for 6 h. The resulting supernatant, adjusted to pH 7.5, was used to resuspend the strain M3B and reacted for 6 h.”

Q26: Line 272 – “After optimizing the concentration of M2B (sumo-GsOMT1^{V125Q}) (Supplementary Fig. 18), 6 mM **12** and 4 mM dopamine could be converted to 2.14 mM (S)-autumnaline with a moderate yield of 53.50% (Fig. 6c), which is 2.25 times higher than that of the cascade reaction using M1G, M2B and M3B as catalysts.” -> I would consider rephrasing this sentence for clarity

A26: Thanks for your advice. We have changed this sentence to “After optimizing the concentration of M2B (sumo-GsOMT1^{V125Q}) (Supplementary Fig. 18), 2.14 mM (S)-autumnaline was produced from 6 mM **12** and 4 mM dopamine with a moderate yield of 53.50% (Fig. 6c). Compared with the cascade reaction catalyzed by strains M1G, M2B (*RnCOMT*) and M3B, the yield of (S)-autumnaline was increased by 2.25 times.” (Please see lines 296-299).

Meanwhile, in order to facilitate understanding, we have defined the strain M2B in the revised manuscript and Supporting Information. The strain M2B expressed protein *RnCOMT* was defined as M2B (*RnCOMT*); the strain M2B expressed protein GsOMT1^{V125Q} was defined as M2B (GsOMT1^{V125Q}) and the strain M2B expressed protein sumo-GsOMT1^{V125Q} was defined as M2B (sumo-GsOMT1^{V125Q}).

Reviewer #2 (Remarks to the Author):

The authors present an interesting biochemical scheme for production of new molecules. There are a few aspects that were not well described in the paper:

A. We would like to thank your professional feedbacks and constructive suggestions for

this manuscript, which help us improve the quality of this manuscript. Here we prepared this point-to-point response and highlighted the changes in blue.

Q1: Availability/synthesis of the PP-acid was not well-described. How will this be conducted (via biological or chemical means)?

A1: Thanks for your question. We have described the source of the PP-acid in the revised manuscript. The substrates **1-11** are commercially available (Supplementary Table 2), and the substrate **12** is obtained by chemical method described in “**Materials and Methods**”. (Line 348-349 and Line 450-457, respectively).

Q2: The overall promiscuity and evolution of the NCS enzyme in the 1st module should be discussed further for varying PP-acids. How promiscuous is this system? How much re-engineering will need to be done to produce other PEIAs? How will this selection take place?

A2: Thanks for your valuable suggestions. We have discussed these questions in the revised manuscript (lines 318-328) as follows:

“Based on the results of this study and previous work (Yang, L. et al. *Chem. Sci.* **11**, 364-371 (2020) and Nishihachijo, M. et al. *Biosci. Biotechnol. Biochem.* **78**, 701-7 (2014)), NCSs were observed to strictly recognize the amine moiety of the substrate and only accept the 3-hydroxyphenyl-2-ethylamine derivatives. However, NCSs could recognize a variety of aldehydes, which is supported by the results of substrate scope investigation. Although NCSs have great promiscuity for various aldehydes, the yields of final products from different aldehydes are affected by the promiscuity of the OMT module (Fig. 5b). Single substituent groups (**2-10**) on the different positions of benzene ring gave good yields of (S)-**2d**–(S)-**10d**, and the yields of di-, tri-substituted products decreased slightly. However, as for the specific substrate **12**, this reaction system did not deliver a satisfactory result. This might be attributed to the similarities between the 6-OMe and 7-OH of (S)-**12c** and the 3'-OH and 4'-OMe of (S)-**12b** (Fig. 5a), which would occupy the binding pocket of RnCOMT to limit the production of (S)-**12c**. As for this kind of substrates, the discovery and engineering of new OMTs will be necessary for achieving high production of the desired

PEIAs.”

Q3: The authors indicate that error bars are duplications, but it is not clear what these truly represent (biological or technical)?

A3: Thanks for your professional suggestion. We have changed the sentence to “All data is presented as mean value of three independent experiments and the error bars indicate \pm sd.” in the figure caption of Fig. 2, Fig. 3, Fig. 4 and Fig. 6. (Please see lines: 100-101, 186-187, 234-235, 290-291)

Q4: Where is the remainder of the carbon going in these timecourses? It would be helpful to see other side products as the yields are not pure bioconversion yields here.

A4: Thanks for your constructive comment. As for the remainder of the carbon going in these timecourses, the reason may be that the oxidation of the dopamine by air or its consumption by the microorganisms and a part of (S)-**12c** remaining within the cell pellet due to the high density of M2B (sumo-GsOMT1^{V125Q}). The relative discussion was added in lines 299-304 as follows:

“However, its yield is still low. The reason might involve the following several aspects: the oxidation of the dopamine by air or its consumption by the microorganisms, a part of (S)-**12c** remaining within the cell pellet due to the high density of M2B (sumo-GsOMT1^{V125Q}) and the insufficient catalytic activity of GsOMT1^{V125Q} and CNMT* towards (S)-**12b** and (S)-**12c** (Fig. 6c), respectively. Therefore, in order to increase the titer of (S)-autumnaline, further protein engineering of GsOMT1^{V125Q} and CNMT* may be required.”

Reviewer #3 (Remarks to the Author):

The author demonstrated a *E. coli*-based bioproduction of phenethylisoquinolines fed with pp-acids and dopamine. *E. coli* has natural pathway to convert pp-acids to the corresponding aldehyde. The strategies of using a shunt pathway & structural diversity of fed substrates have been previously utilized and demonstrated in the production of benzylisoquinoline alkaloids in *E. coli*. To enhance the conversion towards the final products by the unnatural substrates, the team has conducted protein engineering to make

the biocatalyst of each step more efficient. The authors have also engineered E. coli to decrease the consumption of the substrate. Overall, they were able to achieve a significant increase in conversion and a very encouraging titer (~ g/L) for all the natural and unnatural phenethylisoquinolines.

There are several major drawbacks that the author needs to address in the discussion:

A: We would like to thank the reviewer for the thoughtful comments and constructive suggestions, which help us improve the quality of this manuscript. Here we prepared this point-to-point response and highlighted the changes in blue.

Q1: it's true that the author was able to detour from the CYPs using fed substrates, but down the pathway towards colchicine for example, there are many more CYPs;

A1: Thanks for your constructive suggestion. According to your advice, we have discussed in the section of **Discussion**. The relative discussion was added in lines 332-336 as follows:

“In this study, the established concise artificial pathway bypasses the reaction catalyzed by P450 CYP75A109, and shortens the biosynthetic steps for the biosynthesis of the colchicine precursor (S)-autumnaline and its derivatives, which could benefit the total biosynthesis of colchicine and its derivatives, or other PEIAs. However, as for the construction of specific skeletons, for instance, the core structure of colchicine, the following modification by P450 or other enzymes may be still required.”

Q2: how much is the fed substrate and a simple economic analysis will be helpful for the audience to appreciate the strategy more;

A2: Thank you very much for your great advice. However, although the substrates **1-12** are commercially or easily access, compounds (S)-**1d**–(S)-**12d** are not commercially available. In addition, as for scale-up or potential industrial preparation, further optimization of the whole process is still required to increase the final titer. Thus, in current case, we are afraid to provide a reasonable economic analysis.

Q3: the high conversion is achieved through a multiple stage whole cell biotransformation, rather than single stage fermentation, which could be time and effort consuming - a discussion on the con and pro of the established process would be helpful for the audience.

A3: Thanks for your suggestion. We have discussed this problem in the revised manuscript (Line 53-55 and lines 328-330) as follows:

Furthermore, combined with the strategy of one-pot multi-step and metabolic engineering, which will circumvent the issues of the burden of protein expression, deficiency of cofactor and side reactions in a single cell, the titer of target products could be improved.

“To further increase the yields of final products, the strategies of enzyme engineering and metabolic engineering could be combined to optimize multi-enzyme cascade, and then effectively address the shortcomings of multi-stage biotransformation, such as several centrifugation and resuspension steps and ATP regeneration.”

Q4: Also, please make sure to proof read the figures and the manuscript, for example, the red line should be 12d instead of 12c in fig 6c.

A4: Thanks for your careful correction. We have thoroughly proofread the figures, the revised manuscript and Supporting Information, checking for spelling, grammar, and formatting. The mistake in Fig. 6c have been corrected as follows:

The red line (S)-**12c** has been changed to (S)-**12d** in Fig. 6c.

Fig. 6. Rational design of GsOMT1 for efficient biosynthesis of (S)-autumnaline.

REVIEWERS' COMMENTS

Reviewer #1 (Remarks to the Author):

Overall, the authors have provided thoughtful responses and changes in response to my prior critiques. I only have a few minor items that should be considered.

- It may be helpful to explicitly define 'catalytic activity' within the manuscript, and not just in the figure captions. As I understand it, the use of 'catalytic activity' means titer? (or in the case of relative activity, the ratio of titers?). If so, it might be more appropriate to just use 'titer' instead. If something else is meant by the use of 'catalytic activity', then could you please state clearly in the text what is being measured/reported, otherwise it is not apparent what is considered for these numbers. (similarly, a short description in the Methods would be useful)
- Fig 4b – please rotate text in the bar graph for readability.
- Line 206 - "Based on the time course (Fig. 4e), M2B (RnCOMT) could convert(S)-1b to (S)-1c with a conversion rate of 92.86% within 5 h after optimizing the concentration of M2B (RnCOMT) strain (Supplementary Fig. 11c, d)."
 - o Is the conversion rate just based upon consumption of (S)-1b? If so, this statement needs to be altered, as (S)-1c is not actually being measured in this experiment.

Reviewer #2 (Remarks to the Author):

The authors have nicely addressed prior comments

Reviewer #3 (Remarks to the Author):

comments addressed

To Reviewer #1 (Remarks to the Author):

Overall, the authors have provided thoughtful responses and changes in response to my prior critiques. I only have a few minor items that should be considered.

A: We would like to thank your professional feedbacks and constructive suggestions for this manuscript again, which further improves the quality of this manuscript. Here we prepared this point-to-point response and highlighted the changes in blue.

Q1: • It may be helpful to explicitly define 'catalytic activity' within the manuscript, and not just in the figure captions. As I understand it, the use of 'catalytic activity' means titer? (or in the case of relative activity, the ratio of titers?). If so, it might be more appropriate to just use 'titer' instead. If something else is meant by the use of 'catalytic activity', then could you please state clearly in the text what is being measured/reported, otherwise it is not apparent what is considered for these numbers. (similarly, a short description in the Methods would be useful)

A1: We apologized for inappropriate description of "catalytic activity". According to your suggestion, we have described "catalytic activity" in the manuscript formatted with highlight (lines 109-110) as follows:

"Here, the catalytic activity of CNMT towards (S)-**1c** was reflected by detecting the titer of (S)-**1d**."

Meanwhile, we have described the calculation method of relative activity in the figure legend of Fig. 2 (lines 538-539) and Fig. 6 (lines 587-588) as follows:

"The relative activity was determined by the ratio of the (S)-**1d** titer of CNMT mutants to that of CNMT^{WT}." in the figure legend of Fig. 2.

"The relative activity was determined by the ratio of the (S)-**12c** titer of GsOMT1 mutants to that of GsOMT1^{WT}." in the figure legend of Fig. 6.

In addition, we have described "catalytic activity" in the section of "**Methods**" (lines

306-307) as follows:

“The catalytic activity of CNMT and GsOMT1 was assessed by detecting the titer of (S)-1d and (S)-12c, respectively.”

Q2: • Fig 4b – please rotate text in the bar graph for readability.

A2: Thank you for your great suggestion. We have rotated text in the bar graph in Fig. 4b as suggested:

Fig. 4. Biosynthesis of (S)-1d in engineered strain IAA.

Q3: • Line 206 - “Based on the time course (Fig. 4e), M2B (*RnCOMT*) could convert(S)-1b to (S)-1c with a conversion rate of 92.86% within 5 h after optimizing the concentration of M2B (*RnCOMT*) strain (Supplementary Fig. 11c, d).”

o Is the conversion rate just based upon consumption of (S)-1b? If so, this statement needs to be altered, as (S)-1c is not actually being measured in this experiment.

A3: Thank you for your constructive comment. The conversion rate of 92.86% is just based upon the consumption of (S)-1b. We have changed the sentence in the manuscript formatted with highlight (line 172) as follows:

“Based on the time course (Fig. 4e), M2B (*RnCOMT*) could convert 92.86% of (S)-1b within 5 h after optimizing the concentration of M2B (*RnCOMT*) strain (Supplementary Fig. 11c, d).”

To Reviewer #2 (Remarks to the Author):

The authors have nicely addressed prior comments

A: We appreciate the reviewer’s comments that have helped enhance the quality of our study.

To Reviewer #3 (Remarks to the Author):

comments addressed

A: Thank you very much again for your excellent comments.